# FINETUNING MoE LLMs WITH CONDENSER EXPERTS

## ABSTRACT

Despite MoE models leading in benchmarks, supervised fine-tuning (SFT) for the MoE architecture remains difficult because its router layers are fragile. Methods such as DenseMixer and ESFT mitigate collapse with dense mixing or auxiliary load-balancing losses, but these introduce noisy gradients that often degrade performance. In preliminary experiments, we systematically removed experts and observed that while certain "super experts" are activated far more frequently, discarding less used experts still leads to notable performance degradation. This suggests that even rarely activated experts encode non-trivial knowledge useful for downstream tasks. Motivated by this, we propose a new auxiliary loss free MoE SFT framework that combines router biases with shared condenser experts. Instead of enforcing balanced activation across all experts, our method leverages bias updates to encourage imbalanced and sparse routing, allowing rarely used experts to become inactive while designating two existing experts as shared condensers that aggregate knowledge from the inactive set without increasing the per-token compute budget. Router stability is maintained entirely through bias updates that regulate token-level and expert-level activation, eliminating the need for auxiliary losses. Experiments on large-scale MoE models demonstrate that our approach outperforms state-of-the-art SFT baselines such as DenseMixer and ESFT, achieving 4%+ gain on both mathematical reasoning and commonsenseQA benchmarks. Pruning and inter-expert correlation analyses confirm that our condenser experts aggregate knowledge from the long-tail experts, preserving performance under sparse routing.

## 1 INTRODUCTION

Mixture-of-Experts (MoE) models scale language models efficiently by activating only a small subset of experts per token, enabling massive capacity without increasing per-token compute. Yet the same sparse routing that drives their success also makes them fragile: MoE relies on a non-differentiable Top-K router, which blocks straightforward gradient flow and makes post-training, such as supervised fine-tuning (SFT), far more difficult than for dense LLMs.

Over the years, researchers have sought to stabilize MoE training through progressively refined routing strategies. GShard (Lepikhin et al., 2020) introduced top-2 gating with heavy auxiliary balancing losses, while Switch Transformers (Fedus et al., 2022) simplified this to a single expert per token. More recent work, such as DeepSeek-MoE (Wang et al., 2024a) and DeepSeek-V3 (Liu et al., 2024a), explored bias-based routers and minimized auxiliary losses to reduce gradient noise and improve efficiency. However, these advances primarily address pre-training. In the post-training setting, SFT remains underexplored: ESFT (Wang et al., 2024b) routes all gradients to the most-activated expert, while DenseMixer (Yao et al., 2025) improves slightly by applying a Straight-Through Estimator (STE) (Bengio et al., 2013) to approximate updates for inactive experts, yet STE introduces biased gradients.

In parallel, recent studies have identified the existence of "super experts"Su et al. (2025) or "super weights"Yu et al. (2025), whose activations dominate the routing and whose removal leads to sharp performance degradation. These findings suggest that a small subset of experts carries disproportionate importance. However, our observations reveal a complementary phenomenon: even the rarely activated experts encode indispensable information, and pruning them also causes substantial performance decline. This highlights the need for fine-tuning strategies that not only preserve the

capacity of frequently activated super experts, but also retain knowledge embedded in the long tail of rarely used experts.

Motivated by these observations, we adapt the bias-updating principle of DeepSeek's Loss-Free Balancing to the post-training setting. Instead of aiming for balanced activation across all experts, we propose an auxiliary-free fine-tuning framework that enforces sparse routing through globally negative biases. This drives rarely used experts toward inactivity, while two designated Condenser Experts stay active and collect gradients from the other experts, effectively consolidating their knowledge. In doing so, our method closes the train–inference routing gap and preserves information from both dominant and rarely activated experts, achieving significant improvements in fine-tuning performance on reasoning and coding benchmarks. Experimental results show that our method consistently outperforms SoTA MoE SFT methods by 4+ points when post-training popular MoE LLMs on commonsense reasoning (PIQA, ARC, SIQA) and math reasoning benchmarks (MATH-500, AIME-25, GPQA, GSM8K, etc). Our implementation is open-source Anonymously [1].

## 1.1 CONTRIBUTIONS

- Through systematic pruning and scaling-law analysis, we show that even rarely activated experts encode indispensable knowledge. Removing them leads to substantial performance degradation, highlighting the need to preserve contributions beyond the most frequently activated "super experts."

- We extend scaling-law analysis to MoE compression by relating performance to the number of expert parameters retained. Our study compares dense merging, expert pruning, and reduced activation budgets, offering new insights into the trade-offs between model size, sparsity, and accuracy.

- We propose **Expert Condenser**, an auxiliary-free fine-tuning framework that enforces sparsity via bias-driven routing while introducing shared Condenser Experts to preserve knowledge from inactive experts. This design narrows the train–inference routing gap and enables stable MoE post-training.

## 2 DOES SAVING "SUPER EXPERTS" MEAN SAVING MODEL PERFORMANCE?

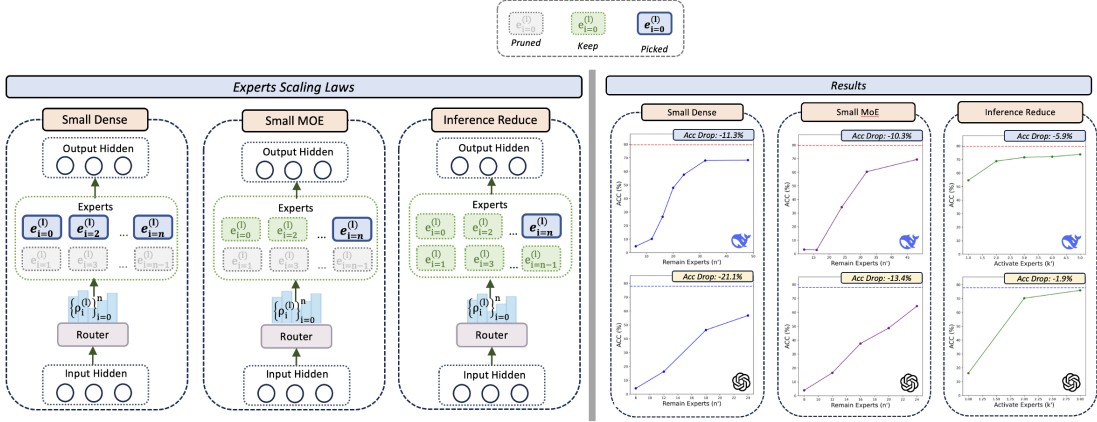

Figure 1: Illustration of the three expert scaling strategies. (*Small-Dense*) Experts are pruned, and all surviving experts are always activated, yielding a smaller dense model. (*Small MoE*) Experts are pruned and only a subset of them are activated per token, resulting in a smaller but still sparse MoE. (*Inference Reduce*) Fewer experts are selected per token while the expert pool is unchanged. Pruned experts are shown in gray, kept experts in light green, and selected experts for a given token in dark blue. A substantial performance gap remains between the base model and pruned variants under Small Dense and small MoE categories.

---

[1] https://anonymous.4open.science/r/Finetuning-MOE-F652

Previous work (Lu et al., 2024; Su et al., 2025) has shown that pruning away frequently activated "super experts" causes large performance degradation, underscoring their importance. However, these studies stop short of asking the complementary question: **is retaining only the super experts sufficient to preserve model quality?**

To answer this, we conduct a systematic scaling-law analysis of experts. Our study takes a more comprehensive view across multiple compression axes blow: (i) Dense Conversion via expert pruning (ii) Smaller MoE conversion via expert pruning, and (iii) reducing activation budget per token comparing to (Jaiswal et al., 2025) focuses specifically on expert pruning within sparse MoE models. Whereas concurrent works (Tian et al., 2025; Nakamura et al., 2025) focus on how the activation ratio (the number of experts active per token) affects accuracy, we instead examine how performance scales with the total number of expert parameters retained. The Top-$k$ selection for token $t$ is defined as

$$S_t = \text{TopK}\big(\{s_{j,t}\}_{j=1}^n, k\big), \qquad g_{i,t} = \mathbf{1}[i \in S_t],$$

where $n$ is the total number of experts, $k$ is the number of experts activated per token, $s_{j,t}$ is the gating score of expert $j$ for token $t$, and $g_{i,t}$ indicates whether expert $i$ is selected[2]. We investigate three strategies as is illustrated in Fig. 1 to study the scaling law by varying $n$ (the size of the expert pool) and $k$ (the activation budget):

(i) **Dense conversion via expert pruning.** We reduce the expert pool from $n$ to $n'$ and activate all surviving experts:

$$n \rightarrow n', \qquad k = n'.$$

$S_t = \{1, \ldots, n'\}$ for all tokens, and the model effectively becomes a smaller dense model.

(ii) **Smaller MoE conversion via expert pruning.** We prune experts from $n$ to $n'$ but keep the activation budget strictly smaller than the remaining pool:

$$n \rightarrow n', \qquad k < n'.$$

The model remains an MoE, since only the top-$k$ experts are selected from the $n'$ survivors.

(iii) **Reducing the activation budget while keeping full model size.** We keep the total number of experts fixed but reduce the activation budget from $k$ to $k'$:

$$n \text{ fixed}, \qquad k \rightarrow k' < k.$$

$S_t$ becomes smaller ($|S_t| = k'$), increasing sparsity while leaving the expert pool unchanged.

To select important experts prior to pruning, we adopt two metrics following ESFT (Wang et al., 2024b): ES-Act (activation ratio) and ES-Mag (weight magnitude). Full definitions are given in Appendix D. Unless otherwise specified, all experiments use ES-Act as the default selection criterion.[3]

To conduct a scaling-law analysis of experts, we design experiments using GPT-OSS OpenAI (2025) and DeepSeek-Coder-V2-Lite (Liu et al., 2024a). In figure 1, we summarize the results across the three strategies. More details of the results are shown in Table 7 and Table 6 in Appendix C. **These results show that saving "Super Experts" is not equivalent to saving model performance.** Although scaling-law trends are evident after pruning, a substantial performance gap remains between the base model and pruned variants. For example, retaining the top 75% of experts still results in more than a 10% drop. Appendix L (Fig. 5) further shows that expert activation is highly skewed: a few "super experts" dominate routing, while a long tail of rarely activated experts—together accounting for only about 10% of activations—still represents a substantial portion of the model's parameter capacity.

Crucially, our experiments reveal that even though a few "super experts" dominate routing, rarely activated experts nonetheless encode indispensable knowledge, and pruning them leads to substantial degradation. This observation motivates our approach: rather than discarding inactive experts, we introduce a condenser-sharing mechanism that aggregates domain knowledge from all the experts.

## 3 PROPOSED METHODS: EXPERT CONDENSER

Our post-training framework addresses a core challenge in MoE: preserving the knowledge distributed across all experts—both dominant and rarely activated—while adapting the model to a new task.

---

[2]Detailed clarifications for all notation are in Appendix A

[3]ES-Mag yields comparable results; see Appendix K for a detailed comparison.

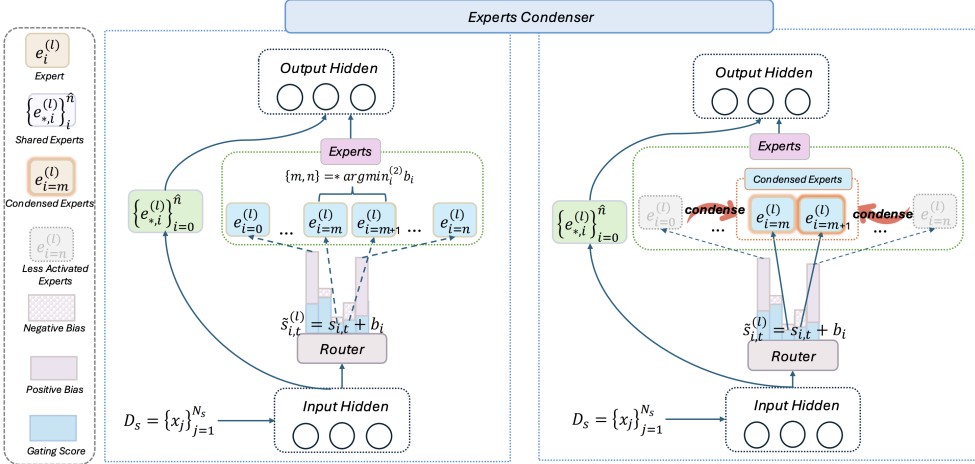

Figure 2: Representation of our Experts Condenser framework. *(Left)* An auxiliary-free router adds trainable biases $b_i$ to logits $s_{i,t}$. Less relevant experts accumulate negative biases, and the two lowest-bias experts $m, n$ are designated as Condenser Experts.*(Right)* These Condenser Experts are always selected during training, ensuring they receive gradients and act as repositories that condense knowledge from inactive experts—preserving information while enforcing sparsity.

Rather than enforcing balanced expert usage, our method explicitly encourages sparsity by down-biasing less relevant experts and systematically transferring their knowledge into two designated **Condenser Experts**.

Our approach has two key components: (i) an auxiliary-free routing mechanism that enforces sparsity through dynamic bias adjustments, and (ii) two always-active Condenser Experts that serve as repositories for aggregated knowledge from inactive experts.

### 3.1 AUXILIARY FREE SPARSITY ENFORCEMENT

A critical challenge in MoE post training is the noise introduced by auxiliary balancing losses. While such losses encourage expert diversity during pre-training, they also inject competing gradient signals that can hinder convergence on specialized downstream tasks [4].

We eliminate this issue by adopting an **auxiliary-loss-free** routing strategy for fine-tuning, where the routing logits are directly modified with trainable bias parameters $b_i{}_{i=1}^n$, one per expert. The router's Top-$k$ selection, $S$, is performed on these biased logits: $S = \text{TopK}\left(\{s_i + b_i\}_{i=1}^n\right)$ where $s_i$ is the original raw logit for expert $i$. The crucial distinction is that the final gating weights, $\rho_i$, are still computed from the original unbiased logits $s_i$, isolating the routing decision from the output computation.

Unlike pre-training methods that adjust biases to *prevent* routing collapse [5], our objective in fine-tuning is to *induce* a controlled collapse. Biases for experts that are rarely useful for the target task are progressively decreased, making them unlikely to be selected. This naturally separates experts into two groups: a small set of task-relevant "active" experts and a long tail of "inactive" experts. By explicitly enforcing sparsity, this mechanism narrows the train–inference routing gap and lays the foundation for our condenser strategy.

### 3.2 HYBRID SHARED EXPERTS WITH GUARANTEED GATING

Our architecture employs a new "shared" computation, which introduces a distinct type of condenser experts in addition to traditional ungated shared experts [6], as visually referenced in Fig. 2.

---

[4] More explanation about auxiliary losses can be found in Appendix E

[5] More details about utilizing auxiliary-loss-free biases to prevent loading unbalance are in Appendix F

[6] The theoretical rationale behind the design of the condenser experts is provided in Appendix O.

**Traditional type-G (Ungated) Shared Experts.** The traditional (ungated) shared experts, shown as *green* boxes ($\left\{e_{*,i}^{(l)}\right\}_{i=0}^{\hat{n}}$), is a set of "Type-G" Shared Experts. These behave as standard feed-forward layers that are applied to every input token $x_j$. Unlike routed experts, they do not receive gating weights; instead, their outputs are summed directly into the representation: $h^{(g)} = \sum_{i=1}^{\hat{n}} \text{FFN}_i^{(G)}(x_j)$

**Proposed type-B (Guaranteed Gated) Shared Expert.** We propose *Type-B* condenser Shared Experts, shown as *blue* boxes drawn from the routed expert pool ($\left\{e_i^{(l)}\right\}_{i=0}^{n}$). Let us denote this expert as $\text{FFN}_j^{(B)}$. This expert is "shared" in the sense that it is **guaranteed to be selected** for every token. This selection is performed just once at the start of the fine-tuning process, by identifying the two activated experts with the lowest-bias. Throughout the entire post training procedure, these two experts are then statically enforced into the active set for every token, supplementing the k-2 specialists dynamically chosen by the router.

The routing process is therefore modified: the router selects the Top-$(k-2)$ experts from the remaining $n-2$ blue experts, and these two special experts $\{j\}$ are *always* added to the active set $S$. Thus, $S = \text{TopK}_{i \in \{1..n\} \setminus \{j\}}(\{s_i + b_i\}, k-2) \cup \{j\}$, ensuring $|S| = k$.

The key distinction—why these condenser experts are "different from the green ones"—is that it is **both shared (always selected) and gated**. Like all other $k-2$ selected experts in $S$, it receives a computed gate weight $g_{i,j}$ from the router. $h^{(b)} = \sum_{i \in S} g_{i,j} \text{FFN}_i^{(B)}(x_j)$

The final layer output $h_t'$ combines the residual, Type-G shared path, and Type-B gated path: $h_t' = u_t + h^{(g)} + h^{(b)}$. This method ensures a baseline of common knowledge (from Type-G) while also forcing the model to always utilize and weigh the contribution of a specific, powerful "capillary" expert (Type-B), supplemented by $k-2$ other dynamically chosen specialists.

## 4 BACKGROUND AND RELATED WORKS

Post-training for Mixture-of-Experts (MoE) large language models remains relatively underexplored. Recent efforts have primarily focused on how to adapt experts so that they better align with downstream domains. Two representative approaches are Expert Supervised Fine-Tuning (ESFT) (Wang et al., 2024b) and DenseMixer (Yao et al., 2025), which propose different strategies for handling gradient propagation through the non-differentiable Top-$k$ routing mechanism.

**ESFT** [7] ESFT strengthens the role of the most frequently activated experts by routing gradients only through the Top-$k$ set $S_t$. Formally, if $w_{i,t}$ denotes the routing weight of expert $i$ for token $t$, then the gradient with respect to router parameters $\theta$ is approximated as $\nabla_\theta \mathcal{L} \approx \sum_{i \in S_t} \left( \frac{\partial \mathcal{L}}{\partial y_t} \cdot v_i \right) \frac{\partial w_{i,t}}{\partial \theta}$. Here $S_t = \text{TopK}(\{w_{j,t}\}_{j=1}^n, k)$ is the set of selected experts, and $v_i$ is the output of expert $i$. While this approach explicitly reinforces the "super experts" that dominate activation, the less activated experts are frozen and treated as trivial experts.

**DenseMixer.** [3] DenseMixer instead addresses the non-differentiability of Top-$k$ routing by adopting a straight-through estimator (STE). In this view, the backward pass ignores the hard selection and treats TopK as the identity map: $\frac{\partial \text{TopK}(w_{1,t},...,w_{n,t})_i}{\partial w_{j,t}} \approx \delta_{i,j}$. As a result, gradients flow to *all* experts' routing weights, not just those in $S_t$: $\nabla_\theta \mathcal{L} \approx \sum_{i=1}^n \left( \frac{\partial \mathcal{L}}{\partial y_t} \cdot v_i \right) \frac{\partial w_{i,t}}{\partial \theta}$. However, both ESFT (Wang et al., 2024b) and DenseMixer (Yao et al., 2025) still suffer from the additional noise introduced by auxiliary balancing losses. Moreover, they overlook the fact that less frequently activated experts also encode indispensable knowledge and contribute significantly to overall model performance.

## 5 EXPERIMENTAL RESULTS

**Model Architecture and Dataset:** In our experimental setup, we use open-weight GPT-OSS-20B (OpenAI, 2025), Deepseek-V2-Lite (Liu et al., 2024a), Deepseek-Coder-V2-Lite-Instruct (Liu et al., 2024a), OLMoE-7B-01-25 (Muennighoff et al., 2024), Qwen1.5-MoE-A2.7B (Yang et al., 2024), and Qwen3-30B-A3B (Yang et al., 2025) to conduct experiments.

---

[7]More details about ESFT and DenseMixer are in Appendix G

In Section§ 2, We study the MoE expert scaling law using Deepseek-Coder-V2-Lite-Instruct base model and GPT-OSS-20B base model, and test the evaluation performance on `MultiArith`, `GSM_8K` (Cobbe et al., 2021), `AddSub`, `AQuA`, `SingleEq`, `SVAMP`, and `mawps`.

In Subsection § 5.1, we evaluate MoE post-training algorithms on math reasoning domains. We fine-tune on the `Math7K` and `Math14K` dataset using the DeepSeek-V2-Lite, Qwen1.5-MoE-A2.7B, OLMoE-7B-01-25 and test the evaluation performance on downstream testsets `MultiArith`, `GSM_8K` (Cobbe et al., 2021), `AddSub`, `AQuA`, `SingleEq`, `SVAMP`, and `mawps`. Then, we post-train Qwen1.5-MoE-A2.7B, DeepSeek-Coder-V2-Lite-Instruct, and Qwen3-30B-A3B on `Stanford-S1` dataset (Muennighoff et al., 2025) and test the evaluation performance on downstream SOTA math reasoning benchmarks `AIME2025`, `AIME2024`, `GPQA-Diamond` (Rein et al., 2024), and `MATH-500`.

In Subsection § 5.2, we turn to commonsense reasoning. Following (Hu et al., 2023; He et al., 2024; Liu et al., 2024b), we merge the training sets of eight tasks into `commonsense_15k` and evaluate on their individual test sets: `BoolQ`, `PIQA`, `SIQA`, `HellaSwag`, `ARC-e`, `ARC-c`, and `OBQA`. Results are reported as accuracy, with an averaged score summarizing overall effectiveness. Across all datasets—`Stanford-S1K`, `Commonsense`, `Math7K`, and `Math14K`—our setup emphasizes the generalization ability of LLMs across diverse sub-tasks. In Appendix M, we present ablation studies demonstrating that both components of our approach, the auxiliary-free routing mechanism 3.1 and Condenser Experts 3.2, contribute significantly to the overall performance improvement.

**Training Framework and Hyper-parameters:** We used the `huggingface-trl` (von Werra et al., 2020) library with zero-2 or zero-3 (Ren et al., 2021) for fine-tuning and `vllm` (Kwon et al., 2023), `lighteval` (Habib et al., 2023), and `accelerate` (Gugger et al., 2022) library for inference evaluation. Both training and evaluation are using dtype BF16.

**MoE Post-train Baselines:** For state-of-the-art (SOTA) MoE post-training baselines, we choose to include ESFT (Hu et al., 2021) and DenseMixer (Yao et al., 2025).

**Computational Resources:** We conduct our experiments and implement SOTA baselines of ESFT and DenseMixer Yao et al. (2025) to post-train with 8 NVIDIA H100_80GB GPUs. Communication between the CPU and GPU is facilitated via PCIe-G4 and communication between GPUs is facilitated via Nvlink-3.

## 5.1 MATH REASONING

We evaluate **ExpertCondenser**, our proposed method, against two state-of-the-art MoE post-training approaches: ESFT (Wang et al., 2024b) and DenseMixer (Yao et al., 2025). To ensure a fair comparison, we adopt the same training configurations as prior work, including batch size, data type, learning rate, and sequence length. We re-implemented ESFT and DenseMixer following the reported setups in (Yao et al., 2025).

Table 1 demonstrates that on most SoTA math reasoning benchmarks, ExpertCondenser outperforms baseline methods across Qwen3, DeepSeek-Coder-V2-Lite, and Qwen2. ExpertCondenser enhances accuracy of DenseMixer by 5.9%, 5.3%, and 7.1% on Qwen3, DeepSeek-Coder-V2-Lite, and Qwen2 respectively.

We further reports zero-shot performance after two epochs of fine-tuning on two math reasoning datasets (*Math-7K* [9] and *Math-14K*) in Table 2. Across all benchmarks (GSM8K, SingleEq, SVAMP, MultiArith, AddSub, AQuA, and MAWPS), ExpertCondenser consistently outperforms both ESFT and DenseMixer on DeepSeek-V2-Lite, Qwen2-MoE, and OLMoE. Notably, by more effectively consolidating expert knowledge, ExpertCondenser achieves substantial gains over prior approaches. On *Math-7K*, it improves the average accuracy of Qwen2-MoE from 57.9 (DenseMixer) to 63.4 (**+5.5%**), DeepSeek-V2-Lite from 66.8 to 73.1 (**+6.3%**), and OLMoE from 64.8 to 70.2 (**+5.4%**). On *Math-14K*, ExpertCondenser further boosts performance: Qwen2-MoE rises from 62.9 to 67.9 (**+5.0%**), DeepSeek-V2-Lite from 64.9 to 69.4 (**+4.5%**), and OLMoE from 65.7 to 70.0 (**+4.3%**).

---

[9]More details about `Commonsense` dataset can be found in Appendix J.

Table 1: Evaluation of post-trained models Zero-Shot *P@ss1:4 samples* Results on downstream math reasoning benchmarks after fine-tuning with *Stanford-S1*, including GPQA Diamond, AIME 2024, AIME 2025, and MATH-500.[8]

| Model | Model Size | Activate #Param | Distill Type | GPQA Diamond | AIME 2024 | AIME 2025 | MATH-500 | AVG |
|---|---|---|---|---|---|---|---|---|
| Qwen3 | 30B | 3B | ExpertCondenser(Ours) | 68.8 | 68.3(82/120) | 51.7(62/120) | 96.8 | **71.4** |
| | | | DenseMixer | 61.0 | 65.8(79/120) | 46.7(56/120) | 95.8 | 67.3 |
| | | | ESFT | 52.7 | 61.7(74/120) | 44.2(53/120) | 92.0 | 62.7 |
| | | | SFT | 58.6 | 63.3(76/120) | 48.3(58/120) | 94.8 | 66.3 |
| | | | Base Model | 38.9 | 20.8(25/120) | 7.5(9/120) | 72.6 | 35.0 |
| DeepSeek-Coder-V2-Lite | 16B | 2.4B | ExpertCondenser(Ours) | 40.6 | 9.2(11/120) | 6.7(8/120) | 68.9 | **31.4** |
| | | | DenseMixer | 34.8 | 2.5(3/120) | 2.5(3/120) | 64.8 | 26.1 |
| | | | ESFT | 32.2 | 2.5(3/120) | 2.5(3/120) | 63.0 | 25.0 |
| | | | SFT | 34.2 | 2.5(3/120) | 2.5(3/120) | 64.6 | 26.0 |
| | | | Base Model | 31.9 | 0.8(1/120) | 1.7(2/120) | 62.0 | 24.1 |
| Qwen2 | 14B | 2.7B | ExpertCondenser(Ours) | 34.6 | 6.7(8/120) | 6.7(8/120) | 28.6 | **19.5** |
| | | | DenseMixer | 26.8 | 1.7(2/120) | 0.8(1/120) | 20.4 | 12.4 |
| | | | ESFT | 26.4 | 0.8(1/120) | 0.8(1/120) | 18.1 | 11.5 |
| | | | SFT | 27.8 | 0.8(1/120) | 0.0(0/120) | 20.1 | 12.2 |
| | | | Base Model | 25.9 | 0.0 | 0.0 | 8.4 | 8.6 |

Table 2: Evaluation of post-trained models (Zero-Shot results) on downstream Math Reasoning datasets, including SingleEQ, MultiArith, AddSub, GSM8K, SVAMP, and AQuA.

| Dataset | Model | Model Size | Activate #Param | Post-train Type | GSM8k | SingleEq | SVAMP | MultiArith | AddSub | AQuA | mawps | AVG |
|---|---|---|---|---|---|---|---|---|---|---|---|---|
| math7k | DeepSeek-V2-Lite | 16B | 2.4B | ExpertCondenser (Ours) | 59.4 | 92.5 | 69.1 | 91.5 | 79.5 | 36.1 | 83.6 | **73.1** |
| | | | | DenseMixer | 57.8 | 81.2 | 67.2 | 89.6 | 64.6 | 28.8 | 78.8 | 66.8 |
| | | | | ESFT | 58.6 | 80.9 | 65.8 | 90.7 | 62.3 | 27.6 | 76.1 | 66.0 |
| | | | | SFT | 58.4 | 80.6 | 66.2 | 90.2 | 61.0 | 24.8 | 75.8 | 65.3 |
| | | | | Base Model | 8.0 | 20.0 | 26.6 | 24.0 | 35.4 | 21.4 | 33.6 | 24.2 |
| | QWen2-MoE | 14B | 2.7B | ExpertCondenser (Ours) | 57.2 | 74.6 | 55.7 | 86.0 | 61.8 | 33.1 | 75.6 | **63.4** |
| | | | | DenseMixer | 48.2 | 71.3 | 53.8 | 78.6 | 54.8 | 28.6 | 70.2 | 57.9 |
| | | | | ESFT | 46.9 | 69.3 | 54.1 | 75.7 | 52.2 | 27.6 | 68.1 | 56.2 |
| | | | | SFT | 45.8 | 70.2 | 53.6 | 76.2 | 53.3 | 27.6 | 67.8 | 56.4 |
| | | | | Base Model | 25.6 | 31.3 | 27.4 | 33.5 | 46.8 | 25.4 | 28.2 | 31.2 |
| | OLMoE | 7B | 1B | ExpertCondenser (Ours) | 68.4 | 79.8 | 71.2 | 93.8 | 63.4 | 36.3 | 78.8 | **70.2** |
| | | | | DenseMixer | 64.8 | 78.2 | 92.0 | 56.4 | 58.6 | 30.2 | 73.5 | 64.8 |
| | | | | ESFT | 62.2 | 75.2 | 68.0 | 93.3 | 58.2 | 28.7 | 73.1 | 65.5 |
| | | | | SFT | 63.6 | 74.8 | 67.7 | 92.3 | 57.8 | 27.6 | 72.4 | 65.2 |
| | | | | Base Model | 16.1 | 23.6 | 17.7 | 9.2 | 21.3 | 22.8 | 13.9 | 17.8 |
| | GPT-OSS | 20B | 3.6B | CondenserExperts | 81.7 | 93.2 | 82.5 | 98.5 | 85.6 | 38.6 | 91.6 | **81.7** |
| | | | | DenseMixer | 80.1 | 92.3 | 83.2 | 98.7 | 82.5 | 37.4 | 90.8 | 80.7 |
| | | | | ESFT | 76.6 | 92.9 | 80.2 | 98.2 | 82.0 | 35.4 | 90.3 | 79.4 |
| | | | | Base Model | 77.4 | 82.9 | 84.0 | 91.8 | 79.7 | 31.5 | 92.0 | 77.0 |
| math14k | DeepSeek-V2-Lite | 16B | 2.4B | ExpertCondenser (Ours) | 63.6 | 81.2 | 71.8 | 93.8 | 60.8 | 33.2 | 81.4 | **69.4** |
| | | | | DenseMixer | 59.4 | 78.6 | 67.4 | 89.4 | 57.8 | 28.6 | 73.6 | 64.9 |
| | | | | ESFT | 58.2 | 75.8 | 65.2 | 89.0 | 56.5 | 29.5 | 73.5 | 64.0 |
| | | | | SFT | 57.6 | 76.4 | 67.6 | 90.1 | 59.7 | 30.6 | 74.3 | 65.2 |
| | | | | Base Model | 8.0 | 20.0 | 26.6 | 24.0 | 35.4 | 21.4 | 33.6 | 24.2 |
| | QWen2-MoE | 14B | 2.7B | ExpertCondenser (Ours) | 58.8 | 81.2 | 59.2 | 91.6 | 72.8 | 33.2 | 78.4 | **67.9** |
| | | | | DenseMixer | 52.6 | 75.2 | 56.8 | 87.8 | 65.4 | 28.6 | 73.6 | 62.9 |
| | | | | ESFT | 52.5 | 76.0 | 54.1 | 86.2 | 62.3 | 29.5 | 71.4 | 61.7 |
| | | | | SFT | 51.8 | 74.8 | 55.4 | 87.2 | 66.7 | 27.6 | 74.8 | 62.6 |
| | | | | Base Model | 25.6 | 31.3 | 27.4 | 33.5 | 46.8 | 25.4 | 28.2 | 31.2 |
| | OLMoE | 7B | 1B | ExpertCondenser (Ours) | 67.8 | 81.6 | 72.4 | 86.8 | 68.8 | 32.8 | 79.6 | **70.0** |
| | | | | DenseMixer | 65.8 | 76.8 | 68.2 | 80.6 | 62.4 | 30.7 | 75.3 | 65.7 |
| | | | | ESFT | 64.4 | 77.0 | 68.9 | 81.8 | 64.1 | 30.7 | 74.8 | 65.9 |
| | | | | SFT | 65.3 | 78.4 | 69.8 | 82.8 | 68.5 | 31.4 | 75.8 | 67.4 |
| | | | | Base Model | 16.1 | 23.6 | 17.7 | 9.2 | 21.3 | 22.8 | 13.9 | 17.8 |

## 5.2 OTHER DATASETS

To ensure that our findings above are generalizable, we further examine the performance of ExpertCondenser under the common sense reasoning dataset `CommonSense-15K`, including six downstream test datasets, `BoolQ`, `PIQA`, `SIQA`, `HellaSwag`, `ARC-e`, `ARC-c`, and `OBQA`.

Table 3 reports the performance of DenseMixer, ESFT, and ExpertCondenser on the `CommenSense` dataset[10] We can observe that ExpertCondenser surpasses the DenseMixer by 5.3% on post-trained OLMoE. On post-trained Qwen-2-MoE, ExpertCondenser surpasses the best performance of DenseMixer and ESFT by 3.0% and 3.9%, respectively.

## 5.3 SYSTEM EFFICIENCY UNDER PARAMETER OFFLOADING

Training large-scale MoE models often depends on parameter offloading (e.g., ZeRO-2 and ZeRO-3 (Ren et al., 2021)), where expert weights are dynamically swapped between GPU and CPU to meet memory limits. The efficiency of this process is highly sensitive to activation patterns, as frequent transfers of large expert weights over PCIe or NVLink can dominate runtime.

---

[10]Please note that we are using `CommonSense-15K`, the smaller version of `CommenSense-170K`.

Table 3: Accuracy comparison of OLMoE and Qwen2-MoE with various post-training methods on commonsense reasoning datasets. Results of all ExpertCondenser are obtained using the hyperparameters described in (Liu et al., 2024b) under the same settings.

| Model | Model Size | Post-train method | BoolQ | PIQA | SIQA | HellaSwag | WinoGrande | ARC-e | ARC-c | OBQA | AVG |
|-------|-----------|-------------------|-------|------|------|-----------|------------|-------|-------|------|-----|
| OLMoE | 7B | ExpertCondenser | 66.8 | 79.9 | 72.1 | 80.3 | 78.6 | 86.6 | 70.9 | 75.0 | **76.3** |
| | | DenseMixer | 62.8 | 68.7 | 65.3 | 71.6 | 73.5 | 81.3 | 63.5 | 71.3 | 69.8 |
| | | ESFT | 63.5 | 63.0 | 58.9 | 64.7 | 62.8 | 74.8 | 63.8 | 63.4 | 64.4 |
| | | SFT | 62.5 | 65.8 | 62.8 | 70.7 | 71.4 | 78.4 | 63.7 | 70.6 | 68.2 |
| | | Base Model | 48.9 | 48.3 | 10.9 | 32.6 | 28.7 | 32.9 | 31.4 | 24.8 | 32.3 |
| Qwen-2-MoE | 14B | ExpertCondenser | 72.1 | 84.9 | 75.6 | 81.6 | 79.8 | 88.5 | 78.1 | 84.4 | **80.6** |
| | | DenseMixer | 70.8 | 85.7 | 74.6 | 75.8 | 78.9 | 82.6 | 74.8 | 77.8 | 77.6 |
| | | ESFT | 69.7 | 85.3 | 75.4 | 78.2 | 74.2 | 84.0 | 71.8 | 75.0 | 76.7 |
| | | SFT | 68.8 | 84.7 | 74.5 | 76.8 | 75.6 | 84.6 | 72.8 | 76.4 | 76.8 |
| | | Base Model | 51.0 | 68.1 | 56.2 | 31.0 | 48.3 | 64.8 | 52.3 | 49.2 | 52.6 |

Our Expert Condenser provides a systems advantage by designating two always-active Condenser Experts. Because these experts consistently handle the majority of activations, their parameters can remain resident in GPU memory, avoiding repeated CPU–GPU transfers and reducing offloading overhead. Figure 3(a) shows expert activation counts for DeepSeek-V2-Lite after post-training, where the two Condenser Experts dominate activations and therefore never need to be swapped out of GPU memory.

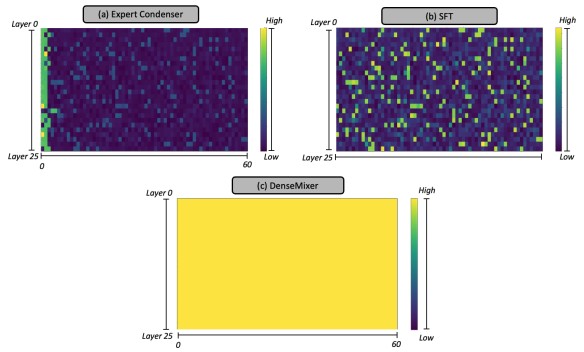

Figure 3: Expert activation counts for all experts across each layer for three methods:(a) Expert Condenser, (b) SFT, and (c) DenseMixer.

Table 4: The experiments involved Post-training, ESFT, DenseMixer, and ExpertCondenser on 8× H100 80GB GPUs using parameter offloading, with a batch size of 32. Communication between the GPU and CPU was facilitated via PCIe-G4.

| DeepSeek-V2-Lite | | | |
|-----------------|-----------------|--------|---------|
| **Post-Train** | **Activate #Params%** | **Time/s** | **Speedup** |
| DenseMixer | 16B | 362.83 | 1× |
| SFT | 2.4B | 152.78 | 2.37× |
| ExpertCondenser | 2.4B | 126.24 | 2.87× |

By contrast, DenseMixer activates *all* experts in each forward pass. This not only introduces extra forward computation but also incurs high cost: every expert's parameters must be loaded into GPU, eliminating sparsity benefits and drastically increasing offloading traffic. This eliminates the computational savings of MoE and dramatically increases offloading traffic, as the system can no longer exploit sparsity to minimize parameter swaps. As shown in Figure 3(c), all experts exhibit uniformly high activation counts, reflecting the full activation pattern. SFT activates only the Top-$k$ experts, saving computation, but the selected set $S_t$ varies across tokens. Figure 3(b) illustrates this behavior: activations are more evenly distributed across experts, but without fixed shared experts, GPU residency is volatile and offloading overhead remains high.

In Table 4, we provide the post-training time costs for DenseMixer, SFT, and ExpertCondenser. ExpertCondenser achieves an 2.87× speedup compared to DenseMixer and outperforms SFT. We conducted time profiling by averaging the post-training time every 10 iterations over 300 iterations, following a 50-iteration warm-up period. Post-training utilized MoE parameter offloading settings to simulate GPU memory limited scenarios. ExpertCondenser offers greater computational efficiency, though these are secondary benefits compared to its primary focus.

## 6 FURTHER ANALYSIS

**Pruning after condensing:** A key question for our framework is whether the two Condenser Experts truly aggregate knowledge from other experts. To evaluate this, we repeat the *dense conversion via ex-*

*pert pruning* experiment[11]. We then compare the performance degradation between ExpertCondenser and ESFT. Table 5 shows the results of pruning while keeping the remaining experts activated. Our method outperforms ESFT by more than 25% across all benchmarks. This confirms the robustness of the always-active Condenser Experts and provides strong empirical evidence that they retain their knowledge during post-training.

Table 5: Evaluation of **DeepSeek-V2-Lite** models after post-training with ESFT and ExpertCondenser, followed by expert pruning.

| Method | Strategies | Remain Experts ($n'$) | Activate Experts ($k'$) | GSM8k | SingleEq | SVAMP | MultiArith | AddSub | AQuA | mawps | AVG |
|---|---|---|---|---|---|---|---|---|---|---|---|
| **ExpertCondenser** | Small Dense | 24 | 24 | 38.6 | 65.3 | 49.3 | 78.7 | 47.6 | 18.1 | 62.4 | **51.4** |
| | Small Dense | 32 | 32 | 49.3 | 73.8 | 56.4 | 81.3 | 53.8 | 23.4 | 68.3 | **58.0** |
| **ESFT** | Small Dense | 24 | 24 | 22.4 | 31.2 | 28.3 | 33.1 | 21.4 | 20.1 | 27.5 | 26.3 |
| | Small Dense | 32 | 32 | 27.4 | 37.6 | 31.4 | 37.2 | 28.8 | 22.4 | 32.5 | 31.6 |

**Expert correlation Analysis:** To assess how fine-tuning alters dependencies between experts, we examine the similarity of their parameter updates. Specifically, we compute the Pearson correlation between the parameters of the condensed experts and those of other experts, comparing the fine-tuned model to the base model. This measure captures linear relationships in parameter changes, allowing us to track how knowledge is redistributed across experts. A formal definition of Pearson correlation is provided in Appendix I.

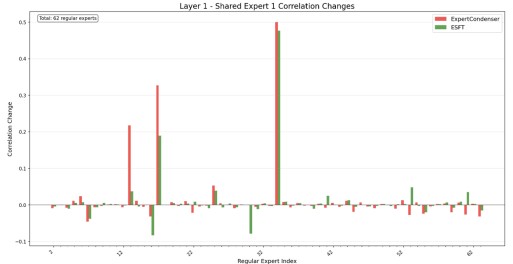
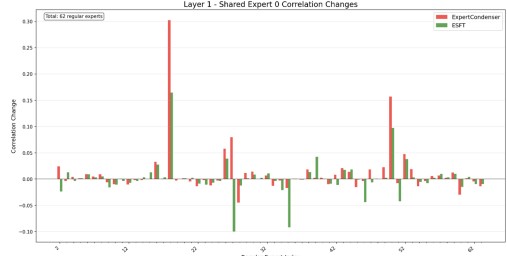

(a) Shared Expert 0 correlation changes.           (b) Shared Expert 1 correlation changes.

Figure 4: Correlation changes between the shared expert and regular experts at Layer 1. Each bar shows how the correlation between a regular expert and the shared expert changes after fine-tuning, compared to the base model. Expert Condenser (red) causes concentrated shifts, markedly strengthening positive correlations while suppressing negative ones. In contrast, ESFT (green) produces smaller, more diffuse adjustments across experts. These results illustrate that Expert Condenser more aggressively reshapes inter-expert relationships, while ESFT exerts a milder influence.

**Observation.** Expert Condenser explicitly targets shared experts and consolidates the capacity of regular experts into the shared expert, resulting in pronounced, concentrated shifts. Regularly routed experts are compared against these shared experts by computing Pearson correlations between their down-projection weight matrices before and after fine-tuning. Relative changes in these correlations—normalized by the base model—reflect how each method reshapes inter-expert dependencies. As shown in Fig. 4, both Expert Condenser and ESFT increase correlations relative to the baseline, with Expert Condenser driving a stronger global increase in correlations. Relative correlation increases average 0.005 more across all layers compared to ESFT. The earliest layer shows the most significant shifts, highlighting Expert Condenser's central role in shaping foundational representations.

# 7 CONCLUSION

Our empirical findings reveal that even rarely activated experts encode indispensable knowledge, and pruning them directly leads to substantial performance degradation. Motivated by this observation, we proposed ExpertCondenser, a post-training framework that leverages bias updates to enforce sparse and imbalanced routing. This design allows rarely used experts to gradually become inactive,

---

[11]Detailed definitions can be found in Section §2. For pruning, we use the ES-ACT metric (see Appendix K) to select the experts to keep, ensuring that the two Condenser Experts are always preserved.

while two designated experts are consistently activated and serve as *Condenser Experts* that aggregate knowledge from the inactive set through backward propagation. By combining sparsity with knowledge preservation, ExpertCondenser significantly outperforms existing post-training methods such as ESFT and DenseMixer across math and commonsense reasoning benchmarks.

ETHICS STATEMENT

This research focuses on the post-training of Mixture-of-Experts (MoE) Large Language Models (LLMs). All datasets used in this work are publicly available and widely adopted in prior research, and all models are open-weighted releases. Our study does not involve human subjects, interventions in live systems, or the use of private or sensitive data. No personally identifiable information (PII) or demographic attributes are included in either the training or evaluation process. As such, we do not identify direct ethical concerns or risks associated with the methodology or findings presented here. Nevertheless, we acknowledge that any advancement in LLM efficiency and performance can indirectly influence downstream applications, and we encourage practitioners to consider the broader societal implications of deploying MoE-based LLMs at scale.

REPRODUCIBILITY STATEMENT

We place strong emphasis on reproducibility and transparency in this work. To enable independent verification, we adopt standardized datasets, provide detailed experimental configurations, and commit to releasing all necessary code and artifacts.

- **Datasets:** All experiments are conducted using open-source and publicly available datasets, ensuring unrestricted access for replication.
- **Algorithms and Models:** We will release the full implementation of our methods, including training and inference scripts, hyperparameter settings, and evaluation protocols.
- **Artifacts:** Preprocessing scripts, simulator code, and pipeline configurations will be made available for end-to-end reproduction of experiments.

Upon camera-ready submission, we will provide a public GitHub repository containing all code and documentation. This repository will enable researchers to reproduce all results, figures, and tables presented in the paper, and to extend our work for future research on MoE post-training.

USE OF LARGE LANGUAGE MODELS (LLMS)

In preparing this project, large language models (OpenAI's GPT-5, Anthropic's Claude) were used for:

- **Editing support**: Suggestions for improving clarity, flow, and conciseness in written sections.
- **Code prototyping**: Assisting with drafting and refining code snippets to test methods and workflows.

All outputs from the model were reviewed, tested, and revised by the author to ensure accuracy and appropriateness for the final submission.

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

# A  NOTATION

For clarity, we summarize the main notations used throughout the theoretical sections.

$h_i$  Raw output vector of expert $i$.

$v_i$  Decomposed expert output: $v_i = \rho_i e_i$.

$\rho_i$  Magnitude (norm) of expert $i$'s output.

$e_i$  Normalized direction vector of expert $i$, $\|e_i\| = 1$.

$y$  Dense output: $y = \sum_{i=1}^{n} v_i$.

$\hat{y}$  Sparse approximation using only $k$ experts.

$S$  Index set of selected experts, $|S| = k$.

$\lambda_i$  Binary indicator of expert selection.

$n, k$  Total number of experts, and number of selected experts.

$v_{i,j,k}^{(l)}$  Output of expert $i$ at layer $l$ for token $k$ in $x_j$.

$g_{i,j,k}^{(l)}$  Gate score assigned to expert $i$ at layer $l$ for token $k$.

$g_{i,t}$  Indicator if expert $i$ is selected for token $t$ (shorthand).

$s_{i,t}$  Gating score (logit) of expert $i$ for token $t$ before normalization.

$w_{i,t}$  Softmax-normalized routing weight assigned to expert $i$ for token $t$, and softmax-normalized routing weight from the *unbiased* scores $s_{i,t}$.

$L_j$  Length (number of tokens) of sample $x_j$.

$\mathcal{D}_s, N_s$  Subset of training data and its size.

$x_j$  $j$th token for training

$s_i^{(l)}$  Magnitude-based expert score (ES-Mag) at layer $l$.

$r_i^{(l)}$  Activation-ratio expert score (ES-Act) at layer $l$.

$K$  Number of experts selected per token.

$S^{(l)}$  Set of selected experts at layer $l$.

$\mathcal{L}_{\text{Balance}}$  Auxiliary loss for balancing expert utilization.

$f_i$  Normalized fraction of tokens routed to expert $i$.

$P_i$  Average routing weight assigned to expert $i$ across tokens.

$\alpha$  Hyperparameter controlling auxiliary loss strength.

$T$  Sequence length (number of tokens).

$\theta$  Router parameters used to compute gating scores.

$\mathcal{L}$  Generic training loss depending on model output $y$.

$b_i$  Expert-wise bias used only for selection to improve load balance.

$\tilde{s}_{i,t}$  Biased gating score for expert $i$ on token $t$: $\tilde{s}_{i,t} = s_{i,t} + b_i$.

$S_t$  Top-$K$ selection set for token $t$ obtained from $\{\tilde{s}_{j,t}\}_{j=1}^{n}$.

$\gamma$  Bias update speed.

## B EXPERIMENTAL SETTINGS

**Training Framework and Hyper-parameters:** We used the `huggingface-trl` (von Werra et al., 2020) library with zero-2 or zero-3 (Ren et al., 2021) for fine-tuning and `vllm` (Kwon et al., 2023), `lighteval` (Habib et al., 2023), and `accelerate` (Gugger et al., 2022) library for inference evaluation. Both training and evaluation are using dtype BF16.

**MoE Post-train Baselines:** For state-of-the-art (SOTA) MoE post-training baselines, we choose to include ESFT (Hu et al., 2021) and DenseMixer (Yao et al., 2025). The number of epochs, learning rate, and batch size will be the same when conducting experiemnt on the same model to ensure a fair comparison across different methods.

**Computational Resources:** We conduct our experiments and implement SOTA baselines of ESFT and DenseMixer Yao et al. (2025) to post-train with 8 NVIDIA H100_80GB GPUs. Communication between the CPU and GPU is facilitated via PCIe-G4 and communication between GPUs is facilitated via Nvlink-3.

## C TABLE RESULTS FOR EXPERTS SCALING LAWS

Tables 6 and 7 report detailed results of our scaling law experiments on two representative MoE models: GPT-OSS-20B and DeepSeek-Coder-V2-Lite. We evaluate performance under three pruning strategies introduced in Section §2:

(1) *Small Dense Conversion*, where the number of experts is reduced from $n$ to $n'$ and all surviving experts are activated ($k = n'$);

(2) *Inference Reduction*, where the total number of experts is fixed ($n' = n$) but the activation budget is reduced from $k$ to $k' < k$; and

(3) *Small MoE Conversion*, where both the total number of experts is reduced ($n' < n$) and the activation budget is kept sparse ($k < n'$).

Table 6: Evaluating base GPT-OSS-20B model Zero-Shot Results on downstream Math Reasoning dataset, includes SingleEQ, MultiArith, AddSub, GSM8K, SVAMP, and AQuA.

| Model | Strategies | Remain Experts ($n'$) | Activate Experts ($k'$) | GSM8k | SingleEq | SVAMP | MultiArith | AddSub | AQuA | mawps | AVG |
|---|---|---|---|---|---|---|---|---|---|---|---|
| | Base Model | 32 | 4 | 78.0 | 84.6 | 84.1 | 92.0 | 81.0 | 33.9 | 92.0 | **77.9** |
| | | 8 | 8 | 2.3 | 1.7 | 1.8 | 2.7 | 1.8 | 16.7 | 2.3 | 4.2 |
| | | 12 | 12 | 4.8 | 17.8 | 14.6 | 18.4 | 18.6 | 22.8 | 17.3 | 16.3 |
| | Small Dense | 18 | 18 | 48.6 | 39.6 | 47.8 | 52.6 | 63.7 | 23.8 | 47.8 | 46.3 |
| | | 24 | 24 | 56.9 | 58.2 | 63.5 | 68.5 | 67.5 | 24.6 | 58.7 | 56.8 |
| | | 32 | 1 | 5.5 | 19.7 | 13.6 | 12.8 | 19.5 | 23.2 | 18.1 | 16.1 |
| | Inference Reduce | 32 | 2 | 70.8 | 74.6 | 76.4 | 88.2 | 75.4 | 29.9 | 76.9 | 70.3 |
| GPT-OSS-20B | | 32 | 3 | 75.1 | 83.9 | 84.3 | 93.3 | 81.3 | 33.1 | 80.7 | 76.0 |
| | | 8 | 4 | 1.6 | 2.2 | 3.2 | 3.2 | 2.0 | 12.6 | 2.5 | 3.9 |
| | | 12 | 4 | 6.4 | 18.7 | 21.8 | 13.0 | 18.2 | 21.7 | 16.0 | 16.5 |
| | Small MoE | 16 | 4 | 20.2 | 43.7 | 48.1 | 43.7 | 41.0 | 24.0 | 42.4 | 37.6 |
| | | 20 | 4 | 32.8 | 56.1 | 59.9 | 58.2 | 52.9 | 25.6 | 55.5 | 48.7 |
| | | 24 | 4 | 49.6 | 72.4 | 74.5 | 83.3 | 72.4 | 29.1 | 70.2 | 64.5 |

Table 7: Evaluating base DeepSeek-Coder-V2-Lite-Instruct model Zero-Shot Results on downstream Math Reasoning dataset, includes SingleEQ, MultiArith, AddSub, GSM8K, SVAMP, and AQuA.

| Model | Strategies | Remain Experts ($n'$) | Activate Experts ($k'$) | GSM8k | SingleEq | SVAMP | MultiArith | AddSub | AQuA | mawps | AVG |
|---|---|---|---|---|---|---|---|---|---|---|---|
| | Base Model | 64 | 6 | 82.6 | 95.1 | 83.6 | 94.3 | 89.6 | 26.4 | 86.6 | **79.7** |
| | | 6 | 6 | 1.4 | 1.0 | 1.7 | 2.8 | 1.8 | 22.0 | 2.1 | 4.7 |
| | | 12 | 12 | 1.7 | 11.0 | 4.5 | 7.2 | 13.2 | 24.0 | 9.7 | 10.2 |
| | | 16 | 16 | 11.8 | 38.2 | 22.6 | 32.0 | 31.1 | 17.7 | 31.9 | 26.5 |
| | Small Dense | 20 | 20 | 26.0 | 63.2 | 46.0 | 67.7 | 56.5 | 17.7 | 58.4 | 47.9 |
| | | 24 | 24 | 36.9 | 75.2 | 58.4 | 76.0 | 70.1 | 20.1 | 66.4 | 57.6 |
| | | 32 | 32 | 47.8 | 83.5 | 71.4 | 88.2 | 81.8 | 24.4 | 79.6 | 68.1 |
| | | 48 | 48 | 48.6 | 82.7 | 72.4 | 87.6 | 82.6 | 24.7 | 80.4 | 68.4 |
| DeepSeek-Coder-V2-Lite | | 64 | 1 | 28.5 | 68.7 | 50.7 | 73.0 | 64.6 | 24.0 | 72.6 | 54.6 |
| | | 64 | 2 | 49.2 | 86.6 | 70.5 | 92.8 | 79.2 | 24.8 | 78.2 | 68.8 |
| | Inference Reduce | 64 | 3 | 54.3 | 87.2 | 76.6 | 92.3 | 80.3 | 25.6 | 84.9 | 71.6 |
| | | 64 | 4 | 53.7 | 87.0 | 76.1 | 94.7 | 83.5 | 26.4 | 83.2 | 72.1 |
| | | 64 | 5 | 57.9 | 89.0 | 79.8 | 95.7 | 84.6 | 25.2 | 84.5 | 73.8 |
| | | 12 | 6 | 0.7 | 0.4 | 1.5 | 0.8 | 0.0 | 16.9 | 1.3 | 3.1 |
| | | 16 | 6 | 0.8 | 0.2 | 1.4 | 1.2 | 1.3 | 13.4 | 1.3 | 2.8 |
| | Small MoE | 24 | 6 | 11.3 | 45.9 | 33.5 | 41.8 | 46.6 | 17.7 | 43.3 | 34.3 |
| | | 32 | 6 | 35.6 | 76.4 | 63.4 | 80.7 | 69.7 | 22.0 | 74.8 | 60.4 |
| | | 48 | 6 | 49.0 | 85.4 | 75.2 | 93.2 | 80.8 | 21.7 | 80.7 | 69.4 |

# D  SELECTING TOP-$k$ EXPERTS

We define two top-$k$ selection rules, selecting by magnitude score and selecting by activation ratio. Let $v_{i,j,k}^{(l)}$ be the output of expert $i$ at layer $l$ for token $k$ in sample $x_j$, with gate score $g_{i,j,k}^{(l)}$. Each sample has length $L_j$, and we draw a subset $\mathcal{D}_s = \{x_j\}_{j=1}^{N_s}$ from the training set. We compute a per-expert relevance score and pick Top-$k$ experts for routing or distillation.

**Magnitude Score (ES-Mag).**  Estimate expert importance by average output magnitude:

$$s_i^{(l)} = \frac{1}{N_s} \sum_{j=1}^{N_s} \frac{1}{L_j} \sum_{k=1}^{L_j} \left\| v_{i,j,k}^{(l)} \right\|.$$

When only a scalar amplitude $\rho_{i,j,k}^{(l)}$ is available, we approximate $\|v_{i,j,k}^{(l)}\| \approx \rho_{i,j,k}^{(l)}$; this criterion favors experts with larger norm contributions (see Appendix H for justification).

**Activation Ratio (ES-Act).**  Estimate importance by how often the expert is selected:

$$r_i^{(l)} = \frac{1}{N_s} \sum_{j=1}^{N_s} \frac{1}{L_j} \sum_{k=1}^{L_j} \frac{\mathbf{1}\left[ g_{i,j,k}^{(l)} > 0 \right]}{K},$$

where $K$ is the number of experts selected per token. This captures routing preference and data alignment.

**Selection.**  For layer $l$, choose

$$S^{(l)} = \mathrm{TopK}\big(\{s_i^{(l)}\}_i\big) \quad \text{or} \quad S^{(l)} = \mathrm{TopK}\big(\{r_i^{(l)}\}_i\big),$$

and select the variant by validation performance: *ES-Mag* emphasizes magnitude-dominant contribution, while *ES-Act* reflects gate-driven frequency.

# E  AUXILIARY LOSS FOR LOAD BALANCING

Uncontrolled routing strategies in Mixture-of-Experts (MoE) models often suffer from load imbalance. This manifests in two ways: (i) *routing collapse*, where only a small subset of experts are consistently selected, leading to undertraining of the remaining experts; and (ii) *computational imbalance*, where uneven routing across devices increases latency and reduces efficiency. To mitigate these issues, an auxiliary loss is commonly introduced in SOTA MoE models (Liu et al., 2024a; Wang et al., 2024b).

For a sequence of length $T$, the auxiliary loss is defined as

$$\mathcal{L}_{\text{Balance}} = \alpha \sum_{i=1}^{n} f_i P_i,$$

where $\alpha$ is a hyperparameter controlling the strength of the regularization. Here,

$$f_i = \frac{n}{KT} \sum_{t=1}^{T} \mathbf{1}\left[g_{i,t} > 0\right], \qquad P_i = \frac{1}{T} \sum_{t=1}^{T} s_{i,t}.$$

The term $f_i$ measures the fraction of tokens routed to expert $i$, normalized by the total number of tokens $T$, experts $n$, and the per-token selection budget $K$. The term $P_i$ is the average routing probability assigned to expert $i$, where $s_{i,t}$ denotes the gating score of expert $i$ for token $t$. The loss encourages alignment between routing frequency ($f_i$) and gating probability ($P_i$), thereby preventing collapse and promoting balanced utilization of experts.

# F  AUXILIARY-LOSS-FREE LOAD BALANCING STRATEGY

To improve load balance without introducing an additional loss term, Deepseek-V3 (Liu et al., 2024a) adjust the *selection* rule by adding an expert-wise bias to the gating scores. Let $s_{i,t}$ be the (un-normalized, before Softmax) gating score for expert $i$ on token $t$. We define a biased score

$$\tilde{s}_{i,t} \;=\; s_{i,t} + b_i,$$

where $b_i$ is an expert-specific bias that is updated by a balancing controller (e.g., based on utilization statistics; see remark below). The Top-$k$ *selection set* for token $t$ is then

$$S_t \;=\; \mathrm{TopK}\big(\{\tilde{s}_{j,t}\}_{j=1}^n,\, K\big), \qquad g_{i,t} \;=\; \mathbf{1}[\,i \in S_t\,].$$

**Important distinction (selection vs. weighting).**  The bias $b_i$ is *only* used to influence which experts enter $S_t$. It does *not* modify the routing weights used to combine expert outputs. Weights are obtained from the *unbiased* scores via softmax:

$$w_{i,t} \;=\; \frac{\exp(s_{i,t})}{\sum_{j=1}^n \exp(s_{j,t})},$$

and the token-level MoE output is

$$y_t \;=\; \sum_{i=1}^n g_{i,t}\, w_{i,t}\, v_i \;=\; \sum_{i \in S_t} w_{i,t}\, v_i, \quad \text{with} \quad v_i = \rho_i e_i.$$

Thus $b_i$ affects *who is selected* but never changes the weights $w_{i,t}$ applied to the selected experts in the forward pass.

**Algorithm (per token $t$).**

1. Compute unbiased scores $\{s_{i,t}\}_{i=1}^n$ and weights $w_{i,t} = \mathrm{softmax}(s_{i,t})$.
2. Form biased scores $\tilde{s}_{i,t} = s_{i,t} + b_i$ and select $S_t = \mathrm{TopK}(\{\tilde{s}_{j,t}\}_{j=1}^n, K)$.
3. Set indicators $g_{i,t} = \mathbf{1}[i \in S_t]$ and compute $y_t = \sum_{i \in S_t} w_{i,t} v_i$.

**Bias updating $b_i$.**  Any load-balancing controller can be used to update the biases; for example, one may adjust $b_i$ as a function of the observed utilization $f_i$ and target utilization $K/n$ (e.g., with a moving-average estimator). In DeepSeek-V3, during training, they monitoring the expert load on the whole batch of each training step. At the end of each step, we will decrease the bias term by $\gamma$ if its corresponding expert is overloaded, and increase it by $\gamma$ if its corresponding expert is underloaded, where $\gamma$ is a hyper-parameter called bias update speed. Through the dynamic adjustment, DeepSeek-V3 keeps balanced expert load during training, and achieves better performance than models that encourage load balance through pure auxiliary losses.

# G  GRADIENT PROPAGATION THROUGH TOP-$k$ ROUTING

Consider a Mixture-of-Experts layer where the router produces gating scores $\{s_{i,t}\}_{i=1}^n$ for token $t$. These scores are normalized via a softmax to obtain the routing weights

$$w_{i,t} \;=\; \frac{\exp(s_{i,t})}{\sum_{j=1}^n \exp(s_{j,t})}\,, \qquad i = 1, \ldots, n.$$

We then select the top-$k$ experts according to these weights and form the token-level output

$$y_t \;=\; \sum_{i \in S_t} w_{i,t}\, v_i,$$

where $S_t = \mathrm{TopK}\big(\{w_{j,t}\}_{j=1}^n\big)$ is the index set of the $k$ largest weights, and $v_i = \rho_i e_i$ is the contribution of expert $i$ (with $\rho_i$ the magnitude of $h_i$ and $e_i$ its normalized direction). The final loss for this token is $\mathcal{L} = \mathcal{L}(y_t)$.

**Gradient with respect to router parameters.**  Back-propagation through this layer requires differentiating the loss with respect to the router parameters $\theta$:

$$\nabla_\theta \mathcal{L} \;=\; \sum_{i=1}^n \left(\frac{\partial \mathcal{L}}{\partial y_t} \cdot v_i\right) \cdot \frac{\partial\, \mathrm{TopK}(w_{1,t}, \ldots, w_{n,t})_i}{\partial w_{i,t}} \cdot \frac{\partial w_{i,t}}{\partial \theta}.$$

The Jacobian term $\partial w_{i,t}/\partial \theta$ is determined by the softmax of the gating scores $s_{i,t}$, while the middle factor contains the (non-differentiable) Top-$k$ selection.

**Conventional approximation: SFT and ESFT**  A common approximation in MoE training treats the Top-$k$ operation as if it were differentiable by passing gradients only through the selected experts. Formally, one replaces

$$\frac{\partial\, \mathrm{TopK}(w_{1,t}, \ldots, w_{n,t})_i}{\partial w_{j,t}} \;\approx\; \delta_{i,j}\, \mathbf{1}[i \in S_t],$$

where $\delta_{i,j}$ is the Kronecker delta. Under this approximation, the router gradient reduces to

$$\nabla_\theta \mathcal{L} \;\approx\; \sum_{i \in S_t} \left(\frac{\partial \mathcal{L}}{\partial y_t} \cdot v_i\right) \frac{\partial w_{i,t}}{\partial \theta}.$$

Thus only the experts chosen in $S_t$ receive gradient updates through the gating mechanism.

**Straight-through (STE) approximation: DenseMixer**  An alternative, more precise approximation—used in methods like DenseMixer—employs a straight-through estimator (STE). In the backward pass, the Top-$k$ operation is treated as the identity map:

$$\frac{\partial\, \mathrm{TopK}(w_{1,t}, \ldots, w_{n,t})_i}{\partial w_{j,t}} \;\approx\; \delta_{i,j}.$$

This allows gradients to flow to all experts' routing weights, yielding

$$\nabla_\theta \mathcal{L} \;\approx\; \sum_{i=1}^n \left(\frac{\partial \mathcal{L}}{\partial y_t} \cdot v_i\right) \frac{\partial w_{i,t}}{\partial \theta}.$$

In this view, the forward pass still uses a hard Top-$k$ selection, but the backward pass distributes gradients as though the selection were an identity operator.

**Summary.** The conventional method restricts gradient updates to the selected experts $S_t$, while the straight-through method propagates gradients to all experts by overriding the Top-$k$ operation in the backward pass.

## H    THEORETICAL SUPPORTS FOR TOP-K SELECTION.

In a Mixture-of-Experts (MoE) architecture, each expert contributes to the overall output as

$$v_i = \rho_i e_i,$$

where $\rho_i$ is the gating weight corresponds to the $i$-th expert, $e_i$ is the expert output, and $v_i$ is the output vector of the $i$-th expert. In a dense model, the final output is given by

$$y = \sum_{i=1}^{n} v_i = \sum_{i=1}^{n} \rho_i e_i.$$

In a sparse Mixture-of-Experts (MoE) model, we aim to reduce computation by selecting only a subset of experts. Thus, we wish to approximate $y$ using

$$\hat{y} = \sum_{i \in S} v_i,$$

where $S$ is a subset of indices with $|S| = k \ll n$. This objective can be formulated as the minimization problem

$$\min_{\lambda_1, \ldots, \lambda_n \in \{0,1\}} \left\| \sum_{i=1}^{n} v_i - \sum_{i=1}^{n} \lambda_i \, v_i \right\|^2 \quad \text{subject to} \quad \sum_{i=1}^{n} \lambda_i = k,$$

where $\lambda_i = 1$ indicates that expert $i$ is selected, and $\lambda_i = 0$ indicates it is omitted. In many practical scenarios, especially when the normalized directions $e_i$ are not strongly correlated, this minimization is well approximated by selecting the experts with the largest values of $\rho_i$. Intuitively, experts with large $\rho_i$ contribute most significantly to the norm of $y$, so preserving these in the approximation yields a smaller error. We analyze why selecting experts with large $\rho_i$ is a reasonable approximation in the following.

In a Mixture-of-Experts (MoE) architecture, each expert contributes to the overall output as

$$v_i = \rho_i e_i,$$

where $\rho_i$ is the gating weight and $e_i$ is the expert output. When analyzing why the top-$k$ selection rule arises, it is instructive to consider two scenarios: one in which the vectors $v_i$ are orthonormal (or nearly so) and another in which they have general correlations.

In this appendix, we show that in the non-orthonormal case, selecting the top-$k$ experts with the largest $\rho_i$ provides a close approximation to the full model output while substantially reducing computational cost. In the orthonormal case, this selection is provably optimal; in the general case, it serves as a widely used and effective heuristic.

### THE ORTHONORMAL (OR WEAKLY-CORRELATED) CASE

Assume that the vectors $v_1, v_2, \ldots, v_n$ are strictly orthonormal, i.e.,

$$v_i^\top v_j = \begin{cases} 0, & \text{if } i \neq j, \\ \|v_i\|^2, & \text{if } i = j. \end{cases}$$

Then, the squared norm of the omitted portion,

$$\left\| \sum_{i=1}^{n} (1 - \lambda_i) v_i \right\|^2,$$

expands as

$$\left\| \sum_{i=1}^{n} (1 - \lambda_i) v_i \right\|^2 = \sum_{i=1}^{n} (1 - \lambda_i)^2 \|v_i\|^2,$$

and since $\lambda_i \in \{0, 1\}$, we have $(1 - \lambda_i)^2 = (1 - \lambda_i)$. Therefore, the objective becomes

$$\sum_{i=1}^{n} (1 - \lambda_i) \|v_i\|^2,$$

subject to $\sum_{i=1}^{n} \lambda_i = k$. To minimize this quantity, it is optimal to set $\lambda_i = 1$ for the $k$ vectors with the largest norms $\|v_i\|^2$ and $\lambda_i = 0$ for the others. In the orthonormal case, this strategy is provably optimal.

Even if the vectors are only weakly correlated, the same principle generally holds: larger magnitudes imply a larger contribution to the overall sum, so omitting vectors with small $\|v_i\|$ results in a minor error, making the top-$k$ selection by magnitude a robust heuristic.

### THE GENERAL (NON-ORTHONORMAL) CASE

When the vectors $v_i$ have significant correlations, the cross terms do not vanish. In this case, the error term becomes

$$\left\| \sum_{i=1}^{n} (1-\lambda_i) v_i \right\|^2 = \sum_{i=1}^{n} (1-\lambda_i) \|v_i\|^2 + 2 \sum_{1 \leq i < j \leq n} (1-\lambda_i)(1-\lambda_j) v_i^\top v_j.$$

Here, the cross terms $v_i^\top v_j$ can affect the error significantly. In principle, finding the subset $S$ that minimizes this expression exactly is an NP-hard combinatorial problem. However, in practice, one commonly uses the heuristic of selecting the top $k$ experts based on the individual magnitudes $\|v_i\|$ (or a predicted magnitude $\rho_i$). This approach is effective because, in many settings, the largest magnitude vectors still dominate the overall contribution even when correlations are present. In scenarios where two high-magnitude vectors are strongly correlated, more sophisticated selection methods might improve the approximation, but the top-$k$ rule remains a strong and computationally efficient baseline.

**Conclusion:** Whether the expert output vectors are orthonormal or generally correlated, the top-$k$ selection rule emerges from the objective of preserving the dominant contributions to the sum while minimizing approximation error. In an MoE architecture, each expert's output $v_i = \rho_i e_i$ contributes to the overall sum. By selecting the $k$ experts with the largest $\rho_i$, one can achieve a good approximation of the full model output with significantly reduced computational cost. In the orthonormal case, this method is exactly optimal, while in the general case it remains a widely-used and effective heuristic.

# I  DEFINITION OF CORRELATION

**Correlation on `down_proj` Weights.**    Consider an MoE layer with a set of experts

$$\mathcal{E} = \{1, \ldots, E\},$$

partitioned into a *shared expert set* $\mathcal{E}_{\text{sh}}$ and a *regular expert set* $\mathcal{E}_{\text{reg}}$, with $\mathcal{E}_{\text{sh}} \cap \mathcal{E}_{\text{reg}} = \emptyset$, $\mathcal{E}_{\text{sh}} \cup \mathcal{E}_{\text{reg}} = \mathcal{E}$.

For expert $e \in \mathcal{E}$ under setting $s$, let

$$\mathbf{w}_{e,s}^{(l)} = \text{vec}\left(W_{\text{down},s}^{(l,e)}\right) \in \mathbb{R}^d,$$

and denote its sample mean by $\bar{w}_{e,s}^{(l)} = \frac{1}{d} \sum_{k=1}^{d} (\mathbf{w}_{e,s}^{(l)})_k$. Define the centered weight vector

$$\tilde{\mathbf{w}}_{e,s}^{(l)} = \mathbf{w}_{e,s}^{(l)} - \bar{w}_{e,s}^{(l)} \mathbf{1}.$$

The Pearson correlation between experts $i$ and $j$ in layer $l$ under setting $s$ is then

$$\mathcal{C}_{ij,s}^{(l)} = \frac{\left\langle \tilde{\mathbf{w}}_{i,s}^{(l)}, \tilde{\mathbf{w}}_{j,s}^{(l)} \right\rangle}{\left\| \tilde{\mathbf{w}}_{i,s}^{(l)} \right\|_2 \left\| \tilde{\mathbf{w}}_{j,s}^{(l)} \right\|_2}.$$

We study three settings:

$$s = 1 : \text{Base}, \qquad s = 2 : \text{ESFT}, \qquad s = 3 : \text{ExpertCondenser}.$$

**Percent Correlation Gain.**    For each shared expert $i \in \mathcal{E}_{\text{sh}}$ and each regular expert $j \in \mathcal{E}_{\text{reg}}$, the percent change in correlation relative to Base is defined as

$$\Delta_{ij,s}^{(l)}(\%) = 100 \frac{\mathcal{C}_{ij,s}^{(l)} - \mathcal{C}_{ij,1}^{(l)}}{\left| \mathcal{C}_{ij,1}^{(l)} \right|}, \qquad s \in \{2, 3\}.$$

This "relative effect size" formulation stabilizes interpretation when the Base correlation is negative or near zero.

**Layer-Level Aggregation.**    The average correlation gain for setting $s$ in layer $l$ is

$$\bar{\Delta}_s^{(l)}(\%) = \frac{1}{|\mathcal{E}_{\text{sh}}| \cdot |\mathcal{E}_{\text{reg}}|} \sum_{i \in \mathcal{E}_{\text{sh}}} \sum_{j \in \mathcal{E}_{\text{reg}}} \Delta_{ij,s}^{(l)}(\%).$$

**Model-Level Summary Across $L$ Layers.**    Define the overall average and variability as

$$\bar{\Delta}_s(\%) = \frac{1}{L} \sum_{l=1}^{L} \bar{\Delta}_s^{(l)}(\%), \qquad \sigma_s = \text{Std}\left(\left\{ \bar{\Delta}_s^{(l)}(\%) \right\}_{l=1}^{L}\right).$$

These summarize how strongly fine-tuning reshapes global correlations between shared experts and regular experts across the entire model.

## J  MATH7K DATASET

`Math10K` dataset can evaluate the effectiveness of LLMs on the arithmetic reasoning task. `Math10K` incorporate six subsets including `GSM8k`, `SingleEq`, `SVAMP`, `MultiArith`, `AddSub`, and `AQuA`.(1) the `GSM8K` (Cobbe et al., 2021) dataset consists of high quality linguistically diverse grade school math word problems created by human problem writers, (2) the `SVAMP` (Patel et al., 2021) benchmark consists of one-unknown arithmetic word problems for up-to-4 grade level students by making simple changes to a set of problems from another existing dataset, (3) the `MultiArith` (Roy & Roth, 2016) dataset of math word problems requiring multiple reasoning steps and operations, (4) the `AddSub` (Hosseini et al., 2014) dataset of addition and subtraction arithmetic word problems, (5) the `AQuA` (Ling et al., 2017) dataset of algebraic word problems with natural language rationales, and (6) the `SingleEq` (Koncel-Kedziorski et al., 2015) dataset of grade-school algebra word problems that map to single equations with varying length;

## K  ES-ACT VERSUS ES-MAG

In this Appendix, we conduct ablation studies to investigate between Magnitude Score(ES-Mag) and Activation Ratio(ES-Act), which one is the better metric to select preserving experts when we converting the original Mixture of Expert model into smaller models (either smaller dense models or smaller MoE models).

Table 8: Evaluation of post-trained models (Zero-Shot results) on downstream Math Reasoning datasets, including SingleEQ, MultiArith, AddSub, GSM8K, SVAMP, and AQuA.

| Metric | Method | Model Size | #Param (Experts) | Post-train Type | GSM8k | SingleEq | SVAMP | MultiArith | AddSub | AQuA | mawps | AVG |
|---|---|---|---|---|---|---|---|---|---|---|---|---|
| ES-Act | Smaller-Dense | -B(6) | - | Base Model | 1.4 | 1.0 | 1.7 | 2.8 | 1.8 | 22.0 | 2.1 | 4.7 |
| | | -B(12) | | | 1.7 | 11.0 | 4.5 | 7.2 | 13.2 | 24.0 | 9.7 | 10.2 |
| | | -B(16) | | | 11.8 | 38.2 | 22.6 | 32.0 | 31.1 | 17.7 | 31.9 | 26.5 |
| | | -B(20) | | | 26.0 | 63.2 | 46.0 | 67.7 | 56.5 | 17.7 | 58.4 | 48.0 |
| | | -B(24) | | | 36.9 | 75.2 | 58.4 | 76.0 | 70.1 | 20.1 | 66.4 | 57.6 |
| | | 8B(32) | | | 47.8 | 83.5 | 71.4 | 88.2 | 81.8 | 24.4 | 79.6 | 68.1 |
| | Smaller-MoE | -B(12) | 2.4B(6) | Base Model | 0.7 | 0.4 | 1.5 | 0.8 | 0.0 | 16.9 | 1.3 | 3.1 |
| | | -B(16) | | | 0.8 | 0.2 | 1.4 | 1.2 | 1.3 | 13.4 | 1.3 | 2.8 |
| | | -B(24) | | | 11.3 | 45.9 | 33.5 | 41.8 | 46.6 | 17.7 | 43.3 | 34.3 |
| | | 8B(32) | | | 35.6 | 76.4 | 63.4 | 80.7 | 69.7 | 22.0 | 74.8 | 60.4 |
| | | -B(48) | | | 49.0 | 85.4 | 75.2 | 93.2 | 80.8 | 21.7 | 80.7 | **69.4** |
| ES-Mag | Smaller-Dense | -B(6) | - | Base Model | 1.6 | 1.2 | 2.1 | 2.6 | 2.1 | 18.9 | 2.3 | 4.4 |
| | | -B(12) | | | 1.8 | 11.8 | 5.2 | 6.8 | 13.6 | 23.8 | 9.4 | 10.3 |
| | | -B(16) | | | 12.6 | 38.8 | 23.8 | 33.8 | 32.6 | 22.4 | 31.9 | 28.0 |
| | | -B(20) | | | 25.4 | 64.8 | 45.6 | 68.8 | 56.3 | 18.2 | 59.8 | 48.4 |
| | | -B(24) | | | 37.4 | 76.8 | 60.2 | 75.4 | 70.8 | 18.8 | 65.2 | 57.8 |
| | | 8B(32) | | | 48.2 | 82.8 | 72.2 | 87.8 | 82.2 | 24.2 | 83.2 | 68.6 |
| | Smaller-MoE | -B(12) | 2.4B(6) | Base Model | 0.7 | 0.4 | 1.7 | 0.6 | 0.0 | 18.6 | 1.6 | 3.4 |
| | | -B(16) | | | 1.3 | 0.4 | 2.6 | 1.8 | 1.6 | 13.6 | 10.4 | 4.5 |
| | | -B(24) | | | 10.8 | 44.7 | 32.8 | 42.3 | 44.8 | 21.3 | 44.8 | 34.5 |
| | | 8B(32) | | | 34.7 | 75.9 | 64.9 | 81.3 | 68.4 | 23.8 | 75.6 | 60.6 |
| | | -B(48) | | | 47.6 | 86.7 | 75.8 | 92.7 | 81.7 | 23.9 | 81.3 | 70.0 |

## L  EXPERT ACTIVATION PROBABILITIES IN BASE DEEPSEEK-V2-LITE

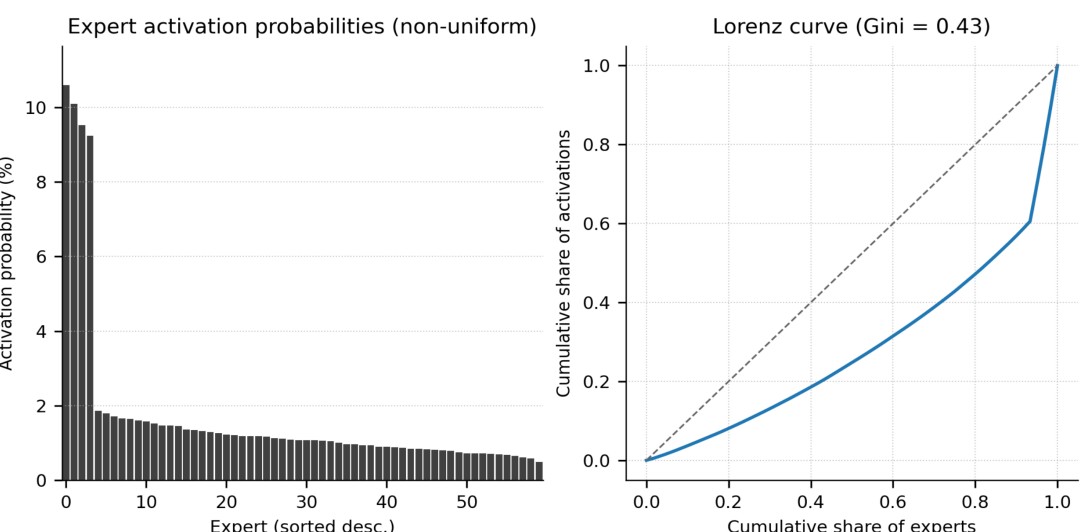

Figure 5: Expert activations rate in whole *math7K* dataset. Expert Activations are tested using the Deepseek-V2-Lite base model.

Figure 5 presents the activation statistics of all experts in the DeepSeek-V2-Lite base model when evaluated on the full *math7K* dataset. The left panel illustrates the sorted activation probabilities and reveals a highly non-uniform routing pattern, where a small subset of experts is used disproportionately often, with several exceeding a 10% activation rate. Beyond this small group, activation frequencies decrease sharply, forming a long tail of under-utilized experts with probabilities falling below 2% and eventually approaching zero. This distribution indicates that, although the MoE architecture allocates equal computational capacity to every expert, the router naturally converges toward a skewed, winner-take-most configuration.

The right panel shows the corresponding Lorenz curve, which characterizes the cumulative distribution of activations across experts. The curve deviates substantially from the diagonal line representing a perfectly uniform distribution, yielding a Gini coefficient of 0.43. This quantitatively confirms a notable degree of inequality in expert utilization, where a relatively small fraction of experts accounts for the majority of routing decisions.

Despite this concentration of activation mass, pruning experiments demonstrate that activation frequency alone is not a reliable indicator of expert importance. Removing the least-active experts leads to clear accuracy degradation, and even when retaining the top 75% of experts ranked by activation probability, the model still suffers a performance drop exceeding 10% on downstream math reasoning benchmarks. Details can be found in Appendix C. This observation suggests that low-activation experts encode specialized competencies that remain essential for certain categories of problems. Consequently, reducing experts solely based on activation sparsity risks eliminating critical functional diversity, thereby motivating the need for structured or distillation-based compression approaches—such as the ExpertCondenser method proposed in this work—to safely reduce mixture-of-experts models.

# M  ABLATION STUDIES

## M.1  DISSECTING THE CONDENSER EXPERT ALGORITHM

In this appendix, we dissect the Condenser Expert algorithm and present empirical results demonstrating that each component of Condenser Experts contributes to the strong post-training performance. Table 18 reports ablation results on both **DeepSeek-V2-Lite (16B)** and **QWen2-MoE (14B)** under the **math7k** benchmark. We evaluate three progressively simplified variants: (i) *aux-free* only, which removes auxiliary losses; (ii) *aux-free+bias*, which additionally incorporates the bias mechanism; and (iii) *aux-free+bias+share*, which further enables expert sharing across tokens.

The results clearly show a consistent trend: performance improves as more components of the Condenser Expert are included. For example, in DeepSeek-V2-Lite, the average score increases from 70.4 (*aux-free*) to 71.2 (*aux-free+bias*) and further to 73.1 when expert sharing is enabled. A similar pattern is observed in QWen2-MoE, where the average accuracy rises from 59.0 to 59.4 and finally to 63.4.

These findings highlight that:

- Removing auxiliary loss alone is not sufficient to stabilize MoE post-training.

- Incorporating bias correction helps mitigate imbalance introduced by sparse optimization.

- Crucially, enabling **expert sharing** provides the largest improvement, indicating that shared experts capture more generalizable knowledge and substantially enhance reasoning performance.

Overall, the ablation validates that each design choice in Condenser Experts is necessary, and that combining all three components yields the best downstream performance.

Table 9: Evaluation of post-trained models (Zero-Shot results) on downstream Math Reasoning datasets, including SingleEQ, MultiArith, AddSub, GSM8K, SVAMP, and AQuA.

| Dataset | Model | Model Size | #Param (Experts) | Post-train Type | GSM8k | SingleEq | SVAMP | MultiArith | AddSub | AQuA | mawps | AVG |
|---|---|---|---|---|---|---|---|---|---|---|---|---|
| math7k | DeepSeek-V2-Lite | 16B | 2.4B | aux-free+bias+share | 59.4 | 92.5 | 69.1 | 91.5 | 79.5 | 36.1 | 83.6 | 73.1 |
| | | | | aux-free+bias | 58.8 | 90.7 | 69.3 | 88.7 | 74.2 | 36.1 | 80.3 | 71.2 |
| | | | | aux-free | 57.6 | 89.6 | 68.6 | 87.5 | 73.8 | 35.2 | 80.4 | 70.4 |
| | QWen2-MoE | 14B | 2.7B | aux-free+bias+share | 57.2 | 74.6 | 55.7 | 86.0 | 61.8 | 33.1 | 75.6 | 63.4 |
| | | | | aux-free+bias | 48.2 | 76.6 | 52.7 | 80.3 | 59.2 | 26.8 | 71.8 | 59.4 |
| | | | | aux-free | 47.2 | 74.0 | 51.8 | 82.0 | 58.7 | 30.3 | 71.8 | 59.0 |

## M.2  HOW TO CHOOSE SHARE EXPERTS

In this subsection, we conduct an ablation study to investigate how experts should be selected as shared experts during post-training. Table 18 reports the results of comparing two selection strategies: (i) choosing high-bias experts and (ii) choosing low-bias experts. Across both **DeepSeek-V2-Lite (16B)** and **QWen2-MoE (14B)**, we observe that selecting *low-bias experts* consistently leads to stronger downstream performance on math reasoning benchmarks. For example, in the **math7k** setting, low-bias experts achieve higher average accuracy (73.1 vs. 72.4 for DeepSeek-V2-Lite and 63.4 vs. 61.8 for QWen2-MoE). These results suggest that low-bias experts encode more generalizable knowledge, making them more effective as shared experts in MoE post-training.

Table 10: Evaluation of post-trained models (Zero-Shot results) on downstream Math Reasoning datasets to conduct ablation study on expert selection based on bias, including SingleEQ, MultiArith, AddSub, GSM8K, SVAMP, and AQuA.

| Dataset | Model | Model Size | #Param (Experts) | Post-train Type | GSM8k | SingleEq | SVAMP | MultiArith | AddSub | AQuA | mawps | AVG |
|---|---|---|---|---|---|---|---|---|---|---|---|---|
| math7k | DeepSeek-V2-Lite | 16B | 2.4B | high bias experts | 60.1 | 90.2 | 70.4 | 90.2 | 74.2 | 37.0 | 84.5 | 72.4 |
| | | | | low bias experts | 59.4 | 92.5 | 69.1 | 91.5 | 79.5 | 36.1 | 83.6 | 73.1 |
| | QWen2-MoE | 14B | 2.7B | high-bias experts | 54.7 | 71.0 | 53.8 | 84.6 | 58.8 | 32.8 | 76.8 | 61.8 |
| | | | | low-bias experts | 57.2 | 74.6 | 55.7 | 86.0 | 61.8 | 33.1 | 75.6 | 63.4 |

## N    TRAINING STABILITY ANALYSIS OF EXPERTCONDENSER

Figure 6 presents the training dynamics of EXPERTCONDENSER compared with the ESFT and DenseMixer baselines when post-training the GPT-OSS model on the *math7K* dataset. The left panel plots the training-loss curves, while the right panel reports the corresponding gradient norms over the full optimization trajectory.

Across training, EXPERTCONDENSER displays markedly improved stability and convergence efficiency relative to the baselines. Its loss curve decreases rapidly during early optimization and consistently settles at a lower final value than both ESFT and DenseMixer. This indicates more effective optimization dynamics and better alignment with the target reasoning distribution. ESFT converges to a higher asymptotic loss, whereas DenseMixer converges slower.

The gradient-norm behavior further highlights this stability advantage. After the initial warm-up phase, EXPERTCONDENSER maintains smooth and well-bounded gradients, free of the high-magnitude spikes observed in DenseMixer and the early instability exhibited by ESFT. The absence of gradient bursts suggests that EXPERTCONDENSER operates over a more stable gradient landscape, which contributes to improved convergence and reduced optimization volatility.

Taken together, these results demonstrate that EXPERTCONDENSER not only delivers better downstream accuracy but also enhances the stability and reliability of the training process compared to existing post-training approaches.

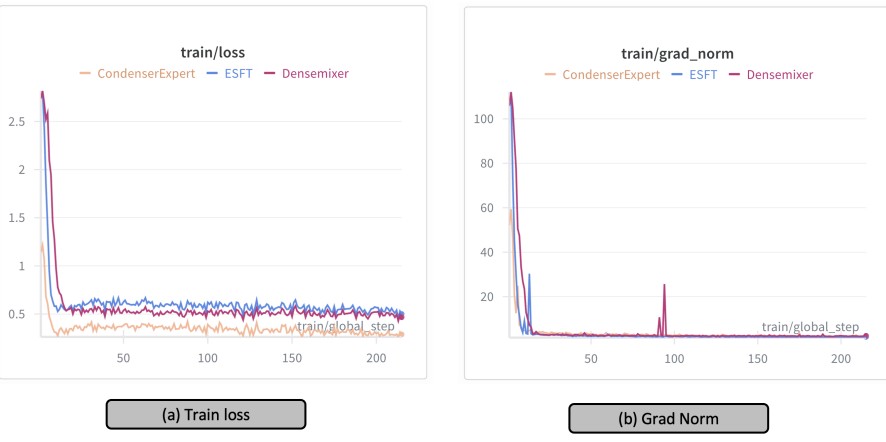

Figure 6: Training stability comparison among EXPERTCONDENSER, ESFT, and DenseMixer. (a) Training loss curves, where EXPERTCONDENSER converges more quickly and reaches a lower final loss. (b) Gradient-norm evolution, showing that EXPERTCONDENSER maintains smooth, stable gradients without the spikes observed in DenseMixer and ESFT.

## O   FORMALIZING THE MOTIVATION BEHIND CONDENSER EXPERTS

Mixture-of-Experts (MoE) architectures commonly rely on two forms of expert capacity: (1) *shared, always-active* experts that provide a global residual function, and (2) *routed, input-adaptive* experts selected by a top-$k$ gating mechanism. In the original DeepSeek-MoE design, the shared experts—referred to as **Type-G (Ungated) Shared Experts**—are applied unconditionally to every token. They contribute a fixed residual transformation,

$$h^{(g)}(x) = \sum_{i=1}^{\hat{n}} \text{FFN}_i^{(G)}(x),$$

independent of router scores or the routing distribution. While this provides constant shared capacity, Type-G experts do not interact with routing dynamics and therefore cannot preserve input-adaptive structure.

To address these limitations, we introduce a new class of experts, called **Type-B (Guaranteed–Gated) Condenser Experts**, which are *always selected* but remain *gated per token*. Let $J_B = \{j_1, j_2\}$ denote the designated condenser experts. For every token $x$, the active set is modified to

$$S(x) = J_B \cup \text{TopK}_{i \notin J_B}(s_i(x),\, k-2),$$

ensuring that $J_B \subset S(x)$ deterministically. The MoE output becomes

$$h^{(b)}(x) = \sum_{i \in S(x)} g_i(x)\, \text{FFN}_i^{(B)}(x),$$

and the condenser experts receive router weights $g_{j_b}(x)$ rather than acting as an unconditional residual. This fundamental change introduces a gated, input-dependent backbone that interacts with the routing mechanism while remaining universally present for every token.

### O.1   INTUITION BEHIND CONDENSER EXPERTS

At a high level, Type-G experts provide a domain-agnostic shared function, whereas our Type-B condenser experts supply a *gated, constrained capacity path* that remains sensitive to the input. The deterministic inclusion of $J_B$ stabilizes router behavior, ensures consistent gradient flow, and prevents collapse in rarely selected experts. This produces a backbone that is simultaneously shared and input-adaptive, enabling the model to consolidate common knowledge while preserving specialization.

### O.2   THEORETICAL PERSPECTIVE

We decompose the output of an MoE layer as

$$h(x) = h_{\text{static}}(x) + h_{\text{B}}(x) + h_{\text{dom}}(x),$$

where $h_{\text{static}}(x) = h^{(g)}(x)$ arises from Type-G shared experts,

$$h_{\text{B}}(x) = \sum_{j \in J_B} g_j(x)\, f_j(x)$$

is the contribution of condenser experts, and

$$h_{\text{dom}}(x) = \sum_{i \in S(x) \setminus J_B} g_i(x)\, f_i(x)$$

captures input-specific domain specialists.

Because $J_B$ is always co-active with domain-routed experts, their gradients encode the joint structure between shared and domain-specific patterns. For tokens drawn from domain $d$, we have

$$\mathbb{E}[\nabla_{\theta_B} \mathcal{L}(x) \mid d] = \mathbb{E}_{x \sim p(x|d)}\left[\frac{\partial \mathcal{L}}{\partial h(x)} \frac{\partial h_{\text{B}}(x)}{\partial \theta_B}\right],$$

implying that condenser experts accumulate gradients that consistently reflect features useful across multiple routed experts. Optimizing over all domains yields

$$\min_{\theta_B} \sum_d \mathbb{E}_{x \sim p(x|d)} \big[ \mathcal{L}(x; \theta_B, \theta_d) \big],$$

which drives $\theta_B$ toward factors that are broadly beneficial across domains. Thus, Type-B experts naturally consolidate domain knowledge without collapsing into a uniform residual, as they are both always present and input-gated.

### O.3 ROUTER STABILITY AND ANTI-COLLAPSE BEHAVIOR

A key benefit of condenser experts is their contribution to router stability. Because their gate logits $s_j(x)$ always participate in the top-$k$ mechanism, router parameters $\phi$ receive non-vanishing gradients:

$$\nabla_\phi \mathcal{L}(x) = \sum_{j \in S(x)} \nabla_\phi g_j(x) \, f_j(x),$$

with $J_B \subset S(x)$ guaranteeing that part of the gradient flows for every token. This effect functions as a structural regularizer: it reduces routing-path variance, prevents long-tail experts from dying, and mitigates the collapse modes seen in sparse MoE routing. Unlike Type-G experts, which bypass gating, condenser experts propagate both forward and backward signals through the router.

### O.4 CAPACITY CONSTRAINTS AND VARIANCE REDUCTION

By enforcing a minimal gated capacity via $J_B \subset S(x)$, the MoE layer incorporates an in-router capacity constraint,

$$\min_\theta \mathbb{E}[\mathcal{L}(x; \theta)] \quad \text{s.t.} \quad J_B \subset S(x), \ |S(x)| = k,$$

which stabilizes the selection process. Under standard stochastic top-$k$ routing, the output variance satisfies

$$\text{Var}[h(x)] = \text{Var}\left[ \sum_{i \in S(x) \setminus J_B} g_i(x) \, f_i(x) \right] + \text{Var}\left[ \sum_{j \in J_B} g_j(x) \, f_j(x) \right],$$

where the second term is router-selection invariant. Condenser experts thus reduce the stochasticity induced by the dynamic expert set, yielding smoother training dynamics.

### O.5 EMPIRICAL EVIDENCE

Across all evaluated benchmarks, replacing ungated shared experts with our gated Type-B condenser experts results in measurable improvements in accuracy, training stability, and expert specialization. The SFT baseline, which relies solely on Type-G experts, consistently underperforms compared with our model augmented with condenser experts (Results are shown in Tab. 1 2 3). These results corroborate the theoretical claims above: enforcing a guaranteed, gated path benefits both the optimization process and the final model capability.

# P HYPER-PARAMETERS

## P.1 MATH7K DATASET AND MATH14K DATASET

In this subsection, we perform an ablation study to verify whether the number of fine-tuning epochs is sufficient for convergence. Since training efficiency and stability are critical in post-training large MoE models, it is important to ensure that extending training does not yield further improvements or lead to overfitting. We therefore evaluate the performance of **DeepSeek-V2-Lite (16B)** and **OLMoE (7B)** under the **math7k** and `math14k` benchmark with **ESFT** fine-tuning for 1, 2, and 3 epochs. The results are reported in Table 12 and 11.

Overall, these findings confirm that our main experiments are conducted with models that have already converged, and that increasing the number of training epochs does not lead to meaningful gains.

Table 11: Evaluation of SFT model Zero-Shot Results on downstream math reasoning tasks after fine-tuning with *Math-14K*, including SingleEQ, MultiArith, AddSub, GSM8K, SVAMP, and AQuA.

| Model | Model Size | #Param (Experts) | Distill Type | GSM8k | SingleEq | SVAMP | MultiArith | AddSub | AQuA | mawps | AVG |
|---|---|---|---|---|---|---|---|---|---|---|---|
| **OLMoE** | **7B** | **1B** | ESFT-1epoch | 55.7 | 76.2 | 58.8 | 71.2 | 62.8 | 28.3 | 64.3 | 59.6 |
| | | | ESFT-2epoch | 52.8 | 78.3 | 59.1 | 71.8 | 63.3 | 29.1 | 68.9 | 60.5 |
| | | | ESFT-3epoch | 52.6 | 77.2 | 57.8 | 72.7 | 64.6 | 31.9 | 70.1 | 60.9 |

Table 12: Evaluation of SFT model Zero-Shot Results on downstream math reasoning tasks after fine-tuning with *Math-7K*, including SingleEQ, MultiArith, AddSub, GSM8K, SVAMP, and AQuA.

| Model | Model Size | #Param (Experts) | Distill Type | GSM8k | SingleEq | SVAMP | MultiArith | AddSub | AQuA | mawps | AVG |
|---|---|---|---|---|---|---|---|---|---|---|---|
| **DeepSeek-V2-Lite** | **16B** | **2.4B** | ESFT-1epoch | 54.1 | 88.0 | 65.3 | 83.7 | 72.7 | 26.8 | 79.4 | 67.1 |
| | | | ESFT-2epoch | 58.6 | 80.9 | 65.8 | 90.7 | 62.3 | 27.6 | 76.1 | 66.0 |
| | | | ESFT-3epoch | 58.2 | 75.8 | 65.2 | 89.0 | 56.5 | 29.5 | 73.5 | 64.0 |
| **OLMoE** | **7B** | **1B** | ESFT-1epoch | 57.0 | 78.5 | 58.6 | 72.0 | 64.3 | 28.3 | 76.1 | 62.1 |
| | | | ESFT-2epoch | 53.8 | 70.9 | 55.7 | 65.0 | 61.5 | 31.5 | 69.7 | 58.3 |
| | | | ESFT-3epoch | 50.3 | 69.3 | 49.5 | 54.5 | 59.2 | 27.6 | 59.2 | 52.8 |

## P.2 GAMMA FOR BIAS UPDATE

Table 13: Evaluation of different post-trained Qwen-2 model Zero-Shot Results on downstream math reasoning tasks with different gamma settings after fine-tuning with *Math-7K*, including SingleEQ, MultiArith, AddSub, GSM8K, SVAMP, and AQuA.

| Model Gamma | Model Size | Distill Type | #Param (Experts) | GSM8k | SingleEq | SVAMP | MultiArith | AddSub | AQuA | mawps | AVG |
|---|---|---|---|---|---|---|---|---|---|---|---|
| G-1e-6 | 14B | aux-free-loss | 2.7B | 48.6 | -76.8 | 50.0 | 81.8 | 59.6 | 29.6 | 70.6 | 59.6 |
| G-1e-5 | 14B | aux-free+bias | 2.7B | 48.8 | 76.4 | 50.2 | 81.6 | 59.8 | 29.7 | 70.8 | 59.6 |
| G-5e-5 | 14B | aux-free+bias | 2.7B | 47.8 | 76.6 | 50.2 | 81.8 | 60.3 | 29.9 | 70.6 | 61.6 |
| G-1e-4 | 14B | aux-free+bias | 2.7B | 48.2 | 76.6 | 52.7 | 80.3 | 59.2 | 26.8 | 71.8 | 59.4 |
| G-1e-3 | 14B | aux-free+bias | 2.7B | 47.7 | 74.2 | 50.6 | 83.2 | 59.0 | 30.7 | 67.2 | 58.9 |
| G-3e-3 | 14B | aux-free+bias | 2.7B | 31.8 | 58.5 | 37.9 | 65.2 | 39.5 | 28.3 | 56.7 | 45.4 |
| G-5e-3 | 14B | aux-free+bias | 2.7B | 15.3 | 32.7 | 26.0 | 47.5 | 29.4 | 21.7 | 33.2 | 29.4 |
| G-1e-2 | 14B | aux-free+bias | 2.7B | 7.2 | 17.3 | 14.7 | 28.2 | 15.7 | 17.7 | 18.5 | 17.0 |

Table 14: Evaluation of different post-trained Deepseek-v2-lite model Zero-Shot Results on downstream math reasoning tasks with different gamma settings after fine-tuning with *Math-7K*, including SingleEQ, MultiArith, AddSub, GSM8K, SVAMP, and AQuA.

| Model Gamma | Model Size | Distill Type | #Param (Experts) | GSM8k | SingleEq | SVAMP | MultiArith | AddSub | AQuA | mawps | AVG |
|---|---|---|---|---|---|---|---|---|---|---|---|
| G-1e-6 | 14B | aux-free-loss | 2.7B | 56.2 | 89.4 | 68.9 | 86.2 | 73.2 | 35.2 | 79.0 | 69.8 |
| G-1e-5 | 14B | aux-free-loss | 2.7B | 56.7 | 89.6 | 69.8 | 87.8 | 74.0 | 35.8 | 79.6 | 70.5 |
| G-5e-5 | 14B | aux-free-loss | 2.7B | 58.6 | 90.6 | 70.2 | 88.6 | 74.4 | 36.2 | 80.6 | 71.1 |
| G-1e-4 | 14B | aux-free-loss | 2.7B | 58.8 | 90.7 | 69.3 | 88.7 | 74.2 | 36.1 | 80.3 | 71.2 |
| G-1e-3 | 14B | aux-free-loss | 2.7B | 43.4 | 78.7 | 62.8 | 80.5 | 60.5 | 25.2 | 71.0 | 60.3 |

# Q  REORGANIZED TABLES FOR MODEL COMPARISONS

The following tables provide detailed comparisons of model performance across different fine-tuning strategies and datasets. Table 15 and Table 16 report zero-shot results on a suite of math reasoning benchmarks (SingleEQ, MultiArith, AddSub, GSM8K, SVAMP, and AQuA) after supervised fine-tuning (SFT) with *Math-14K* and *Math-7K*, respectively. We include both ESFT-tuned variants and their corresponding base models to highlight the effectiveness of expert tuning. Table 17 presents zero-shot pass@1 results (with 4 samples) on more challenging reasoning benchmarks (GPQA Diamond, AIME 2024/2025, and MATH-500) using the *Stanford-S1* dataset. Results are shown for ESFT, DenseMixer, and our proposed ExpertCondenser method, alongside the corresponding base models, enabling direct comparison of different post-training approaches for Mixture-of-Experts LLMs.

Table 15: Evaluation of SFT model Zero-Shot Results on downstream math reasoning tasks after fine-tuning with *Math-14K*, including SingleEQ, MultiArith, AddSub, GSM8K, SVAMP, and AQuA.

| Model | Model Size | Distill Type | #Param (Experts) | GSM8k | SingleEq | SVAMP | MultiArith | AddSub | AQuA | mawps | AVG |
|---|---|---|---|---|---|---|---|---|---|---|---|
| DeepSeek-V2-Lite | 16B | ESFT | 2.4B | 58.6 | 80.9 | 65.8 | 90.7 | 62.3 | 27.6 | 76.1 | 66.0 |
| Qwen2 | 14B | ESFT | 2.7B | 52.5 | 76.0 | 54.1 | 86.2 | 62.3 | 29.5 | 71.4 | 57.1 |
| OLMOE | 7B | ESFT | 1B | 53.8 | 70.9 | 55.7 | 65.0 | 61.5 | 31.5 | 69.7 | 58.3 |
| DeepSeek-V2-Lite | 16B | Base Model | 2.4B | 8.0 | 20.0 | 26.6 | 24.0 | 35.4 | 21.4 | 33.6 | 24.2 |
| Qwen2 | 14B | Base Model | 2.7B | 25.6 | 31.3 | 27.4 | 33.5 | 46.8 | 25.4 | 28.2 | 31.2 |
| GPT-OSS | 20B | Base Model | 3.6B | 77.4 | 82.9 | 84.0 | 91.8 | 79.7 | 31.5 | 92.0 | 77.4 |
| OLMOE | 7B | Base Model | 1B | 16.1 | 23.6 | 17.7 | 9.2 | 21.3 | 22.8 | 13.9 | 17.8 |

Table 16: Evaluation of SFT model Zero-Shot Results on downstream math reasoning tasks after fine-tuning with *Math-7K*, including SingleEQ, MultiArith, AddSub, GSM8K, SVAMP, and AQuA.

| Model | Model Size | Distill Type | #Param (Experts) | GSM8k | SingleEq | SVAMP | MultiArith | AddSub | AQuA | mawps | AVG |
|---|---|---|---|---|---|---|---|---|---|---|---|
| DeepSeek-V2-Lite | 16B | ESFT | 2.4B | 54.7 | 87.2 | 67.3 | 86.8 | 68.1 | 28.7 | 76.3 | **67.0** |
| Qwen2 | 14B | ESFT | 2.7B | 46.9 | 69.3 | 54.1 | 75.7 | 52.2 | 27.6 | 68.1 | 56.2 |
| OLMOE | 7B | ESFT | 1B | 52.8 | 78.3 | 59.1 | 71.8 | 63.3 | 29.1 | 68.9 | 60.5 |
| DeepSeek-V2-Lite | 16B | Base Model | 2.4B | 8.0 | 20.0 | 26.6 | 24.0 | 35.4 | 21.4 | 33.6 | 24.2 |
| Qwen2 | 14B | Base Model | 2.7B | 25.6 | 31.3 | 27.4 | 33.5 | 46.8 | 25.4 | 28.2 | 31.2 |
| GPT-OSS | 20B | Base Model | 3.6B | 77.4 | 82.9 | 84.0 | 91.8 | 79.7 | 31.5 | 92.0 | 77.0 |
| OLMOE | 7B | Base Model | 1B | 16.1 | 23.6 | 17.7 | 9.2 | 21.3 | 22.8 | 13.9 | 17.8 |

Table 17: Evaluation of SFT model Zero-Shot *P@ss1:4 samples* Results on downstream math reasoning benchmarks after fine-tuning with *Stanford-S1*, including GPQA Diamond, AIME 2024, AIME 2025, and MATH-500.

| Model | Model Size | Distill Type | #Param (Experts) | GPQA Diamond | AIME 2024 | AIME 2025 | MATH-500 | AVG |
|---|---|---|---|---|---|---|---|---|
| Qwen3 | 30B | Base Model | 3.3B | 38.9 | 20.6 | 7.7 | 72.8 | 35.0 |
| DeepSeek-V2-Lite | 16B | Base Model | 2.4B | 31.9 | 0.8(1/120) | 1.7(2/120) | 62.0 | 24.1 |
| Qwen2 | 14B | Base Model | 2.7B | 25.9 | 0.0 | 0.0 | 8.4 | 8.6 |
| Qwen3 | 30B | ESFT | 3.3B | 54.8 | 61.6 | 45.6 | 93.4 | 63.9 |
| DeepSeek-V2-Lite | 16B | ESFT | 2.4B | 32.2 | 2.5(3/120) | 2.5(3/120) | 63.0 | 25.0 |
| Qwen2 | 14B | ESFT | 2.7B | 26.4 | 0.8(1/120) | 0.8(1/120) | 18.1 | 11.5 |
| Qwen3 | 30B | DenseMixer | 2.4B | 58.5 | 63.9 | 45.8 | 93.6 | 65.5 |
| DeepSeek-V2-Lite | 16B | DenseMixer | 2.4B | 34.8 | 2.5(3/120) | 2.5(3/120) | 64.8 | 26.1 |
| Qwen2 | 14B | DenseMixer | 2.4B | 26.8 | 1.7(2/120) | 0.8(1/120) | 20.4 | 12.4 |
| Qwen3 | 30B | ExpertCondenser (Ours) | 2.4B | 68.8 | 68.3(82/120) | 51.7(62/120) | 96.8 | 71.4 |
| DeepSeek-V2-Lite | 16B | ExpertCondenser (Ours) | 2.4B | 40.6 | 9.2(11/120) | 6.7(8/120) | 68.9 | 31.4 |
| Qwen2 | 14B | ExpertCondenser (Ours) | 2.4B | 34.6 | 6.7(8/120) | 6.7(8/120) | 28.6 | 19.5 |

# R  GPT-OSS

Table 18: Evaluation of SFT model Results on downstream math reasoning tasks after fine-tuning with *Math-7K*, including SingleEQ, MultiArith, AddSub, GSM8K, SVAMP, and AQuA.

| Model | Model Size | Distill Type | #Param (Experts) | GSM8k | SingleEq | SVAMP | MultiArith | AddSub | AQuA | mawps | AVG |
|---|---|---|---|---|---|---|---|---|---|---|---|
| **GPT-OSS** | **20B** | CondenserExperts | **3.6B** | 81.7 | 93.2 | 82.5 | 98.5 | 85.6 | 38.6 | 91.6 | 81.7 |
| **GPT-OSS** | **20B** | DenseMixer | **3.6B** | 80.1 | 92.3 | 83.2 | 98.7 | 82.5 | 37.4 | 90.8 | 80.7 |
| **GPT-OSS** | **20B** | ESFT | **3.6B** | 76.6 | 92.9 | 80.2 | 98.2 | 82.0 | 35.4 | 90.3 | 79.4 |
| **GPT-OSS** | **20B** | Base Model | **3.6B** | 77.4 | 82.9 | 84.0 | 91.8 | 79.7 | 31.5 | 92.0 | 77.0 |

