# OpenReview forum: "MoECondenser: Finetuning MoE LLMs with Condenser Experts"
_ICLR.cc/2026/Conference — Submitted to ICLR 2026_

### Official Review · Reviewer_cRoC · 2025-10-27

**Soundness:** 3
**Presentation:** 3
**Contribution:** 3
**Rating:** 6
**Confidence:** 4

**Summary:**

This paper proposes Expert Condenser, a new framework for fine-tuning Mixture-of-Experts (MoE) language models without auxiliary losses. The method introduces bias-driven sparse routing combined with two always-active condenser experts that aggregate knowledge from rarely used experts. The method stabilizes fine-tuning by adjusting router biases only.

**Strengths:**

The paper begins with a strong empirical motivation in Section 2: although “super experts” dominate activation in MoE models, even rarely activated experts carry indispensable knowledge. This observation is novel. I believe this finding could be valuable for future improvements of MoE models.

The proposed Expert Condenser introduces bias-driven sparse routing without auxiliary losses, avoiding the gradient noise. I think the method is technically reasonable.

The proposed method outperforms both ESFT and DenseMixer on multiple MoE architectures (Qwen, DeepSeek, OLMoE).

**Weaknesses:**

I think there is a missing link between the analysis in Section 2 and the proposed method in Section 3. It would be better to elaborate on how the findings from Section 2 motivate or inspire the specific design choices made in Section 3.

The bias-update mechanism and condenser aggregation are mainly motivated empirically; the paper lacks formal theoretical grounding or ablation to isolate why condenser experts effectively “absorb” knowledge.

**Questions:**

Are the Type-G and Type-B experts different only in terms of their gating mechanism?

How does the routing strategy of MoE differ between pre-training and post-training settings? It seems that the proposed method is not theoretically connected to SFT. Is this method applicable to the pre-training stage as well?

---

> ### Author Response · Authors · 2025-11-18
>
> Thank you for your valuable review! We would encourage the reviewer to consult our “Response to All Reviewers,” where we list all revisions and additions introduced during rebuttal :)
>
>
>
> > Weakness 1. I think there is a missing link between the analysis in Section 2 and the proposed method in Section 3. It would be better to elaborate on how the findings from Section 2 motivate or inspire the specific design choices made in Section 3.
>
> Thank you for your questions. We appreciate the opportunity to further discuss the purpose of Section 2. We have edited Section 2 to clarify our motivation and improve the presentation. In Section 2, we start from extending scaling-law analysis to MoE compression by relating performance to the number of expert parameters retained. From the dense merging and expert pruning scaling-law analysis, we observe that even though a few “super experts” dominate routing, rarely activated experts nonetheless encode indispensable knowledge, and pruning them leads to substantial degradation (retaining the top 75% of experts still results in more than a 10% drop).
>
> This observation further motivates us: is there a way we can consolidate domain knowledge from those less activated experts, and consistently activate them to further improve MoE post-training performance? Motivated by this, we introduce a condenser sharing experts mechanism that encourages sparsity by down-biasing less relevant experts and systematically transferring their knowledge into designated Condenser Experts.

---

> ### Author Response · Authors · 2025-11-18
>
> >Weakness 2. The bias-update mechanism and condenser aggregation are mainly motivated empirically; the paper lacks formal theoretical grounding or ablation to isolate why condenser experts effectively “absorb” knowledge.
>
> We formularize the theoretical grounding below to further explain why condenser experts can effictively absorb knowledge. More theory of Condenser Experts in Appendix O.
>
> >Our proposed **condenser-experts**, referred to as **Type-B (Guaranteed–Gated)**, are always selected but still gated on a per-token basis. Two experts  $J_B$ = {$j_1, j_2$}   are guaranteed to appear in the active expert set $S(x)$ for every token and receive router weights $g_{j_b} (x)$. The active set becomes:
> >
> >$
> S(x) = J_B \cup\ \mathrm{TopK}_{\,i \notin J_B}\big(s_i(x), k-2\big),
> $
> >
> >and the MoE output is computed as:
> $h^{(b)}(x) = \sum_{i \in S(x)} g_i(x)\,\mathrm{FFN}^{(B)}_i(x)$
> >
> > **How Condenser Experts consolidate domain knowledge:**
> > Enforcing $J_B ⊂ S(x)$ ensures a minimal *input-adaptive* capacity channel that is always active. This provides a guaranteed expert path whose contribution depends on $x$, so domain-relevant features always pass through a gated mechanism rather than a fixed residual.
> >
> > We decompose the MoE layer output as:
> > $h(x) = h_{static}(x) + h_B(x) + h_{dom}(x)$
> >
> > where $h_{static}(x)$ is from ungated shared experts (Type-G),
> >
> > $h_{\text{B}}(x) = \sum_{j \in J_B} g_j(x)\,f_j(x) \)$ is the Type-B path,
> >
> > $h_{\text{dom}}(x) = \sum_{i \in S(x)\setminus J_B} g_i(x)\,f_i(x)$ is the dynamic “domain” expert contribution.
> >
> >Consider a domain label d and tokens sampled from $p(x \mid d)$. The forward router still chooses **domain-specific specialists** in $S(x) \setminus J_B$, while condenser experts are always co-active with them. Under standard gradient-based training, the expected gradient on Type-B parameters $\theta_B$, conditioned on domain d, is:
> >
> >$\mathbb{E}[ \nabla_{\theta_B} L(x) \mid d ]
> = \mathbb{E}_{x \sim p(x \mid d)}[
> (\partial L/\partial h(x))\,(\partial h_B(x)/\partial \theta_B)
> ].$
> >
> > Since $h_B(x)$ always co-occurs with $h_{dom}(x)$, this expectation captures the *joint structure* shared across domain-specific experts. In a multi-domain formulation, condenser experts solve a shared-parameter objective:
> >
> > $min_{θ_B} Σ_d E_{x~p(x|d)} [ L(x; θ_B, θ_d) ]$
> >
> > where $θ_d$ are domain-specific expert parameters in $S(x)$\ $J_B$.
> > Under mild smoothness assumptions, optimization drives $θ_B$ toward features that are consistently useful across domains. Thus, Type-B experts consolidate domain knowledge, maintain sensitivity to the input $x$, and aggregate domain signals from dynamically selected experts—improving domain ability rather than collapsing into a static residual.
>
>
>
> In addition to theory, in Section 6, we also conduct **empirical experiments** to validate condenser can consolidate knowlede. We evaluate this property using two complementary analyses:
>
> >**(1) Pruning-after-Condensing Experiments:** We repeat the dense conversion experiment in Section 2 by pruning experts while always preserving the two condenser experts. If these experts truly aggregate knowledge, the resulting pruned model should retain strong performance despite removing many routed experts.
> As shown in Table 6, models fine-tuned with ExpertCondenser substantially outperform the ESFT-based pruning baseline—exceeding it by over 25% across all downstream math benchmarks. This behavior is only possible if these experts effectively absorb knowledge from the remaining experts during post-training.
> >
> >**(2) Expert Correlation Analysis:** To further analyze how knowledge is redistributed, we compute the Pearson correlation of parameter updates(Details about pearson correlation in appendix I.) between condenser experts and all regular experts, comparing the fine-tuned model against the base model. This measures how closely each routed expert’s updates align with the condenser experts. Figure 7 shows that both ESFT and ExpertCondenser increase correlations relative to the base model, but ExpertCondenser produces a much stronger and more concentrated shift. This indicates that condenser experts serve as a **central aggregation point**, drawing parameter updates from many routed experts. ESFT, in contrast, yields smaller and more diffuse adjustments, suggesting weaker consolidation.
>
> Together, these results demonstrate that the condenser experts absorb and centralize information from other experts. Their robustness under pruning and their dominant correlation patterns both confirm that condensation is occurring in practice.

---

> ### Author Response · Authors · 2025-11-18
>
> > Questions 1. Are the Type-G and Type-B experts different only in terms of their gating mechanism?
>
> We appreciate the chance to discuss this further! While Type-G and Type-B experts differ only in their gating mechanism, this difference leads to different roles and training dynamics.
>
>
>
> For type-G (Ungated Shared Experts), these experts are *always applied* and **never gated**. They act as unconditional residual FFNs:
> $h^{(g)}(x)
> = \sum_{i=1}^{\hat n} \mathrm{FFN}^{(G)}_i(x),$
> providing a fixed shared transformation $f_G(x)$ that does not interact with routing or gradient signals from the gating network.
>
> For type-B (Guaranteed–Gated Experts), two experts $J_B$={$j_1,j_2$} are always selected but always gated per token:
> $
> S(x) = J_B \cup\ \mathrm{TopK}_{\,i \notin J_B}\big(s_i(x), k-2\big),
> $
>
> and the MoE output is computed as:
> $h^{(b)}(x) = \sum_{i \in S(x)} g_i(x)\,\mathrm{FFN}^{(B)}_i(x)$.
> Thus, unlike Type-G, they always participate in both forward and backward routing updates.
>
>
> In short, Type-G provide a global **residual** of shared knowledge. Type-B(the condenser experts) introduce a gated, constrained backbone that: (1) consolidate domain ability, (2) improves router training gradient flow, and (3) stabilizes expert utilization. More formularize theory of Condenser Experts in Appendix O.2~O.4.
>
>
> ---
>
>
> >Question 2. How does the routing strategy of MoE differ between pre-training and post-training settings? It seems that the proposed method is not theoretically connected to SFT. Is this method applicable to the pre-training stage as well?
>
>
> Thank you for the thoughtful question. The routing strategies used in MoE pre-training and post-training differ substantially in both their objectives and their behavior, and this distinction is why our method is designed specifically for the post-training stage.
>
> **Routing in MoE pre-training vs. post-training:**  During large-scale MoE pre-training, the router is typically optimized to *maintain expert diversity*. This is commonly achieved by auxiliary load-balancing losses that penalize routing collapse and ensure that all experts are activated across the large and heterogeneous pre-training corpus. The goal is to encourage each expert to develop different domain specializations and prevent the model from overusing only a few experts. In other words, pre-training routing is designed to spread tokens across experts to maximize domain coverage.
>
> In contrast, the goal in post-training is fundamentally different. Downstream tasks are usually narrow and domain-specific, and forcing expert diversity through an auxiliary objective can introduce conflicting gradients that slow convergence or degrade task performance. In the post-training setting, what we want is not diversity, but task-specific specialization. Our method therefore removes auxiliary losses and introduces a learnable bias adjustment that explicitly *encourages sparsity* in routing. Experts that are not beneficial for the target task are progressively down-biased and naturally become inactive.
>
>
> **Our method is post-training oriented.** Although our method is theoretically compatible with any stage of training, it is specifically motivated by the needs of post-training. Auxiliary-loss-free routing and bias-driven sparsity intentionally encourage the router sparsity, which is desirable when the goal is to sharpen task-specific behavior. For this reason, while we do not run pre-training experiments (as this requires substantial computational resources), we suggest that our framework is not intended for pre-training routing. Our contribution lies in how to enhance and consolidate domain knowledge during downstream adaptation.

---

> ### Author Response · Authors · 2025-11-26
>
> Dear Reviewer cRoC,
>
> Thanks again for the reviews! We deeply appreciate the time and effort you have invested in evaluating our work :)
>
> We've carefully considered your comments and have prepared a detailed rebuttal addressing each point raised. We are hoping reviewers would be able to review our responses and to engage in a discussion before the end of the discussion period. Please let us know if there are any specific areas you'd like us to clarify further or discuss in more detail. We look forward to your thoughts and are eager to engage in a productive dialogue.
>
> Thank you once again for your commitment to the review process.
>
> Best regards,
>
> Authors

---

### Official Review · Reviewer_p9Ex · 2025-11-01

**Soundness:** 3
**Presentation:** 4
**Contribution:** 3
**Rating:** 6
**Confidence:** 2

**Summary:**

Motivated by the observation that even rarely activated experts encode non-trivial knowledge useful for downstream tasks, this paper propose a new auxiliary loss free MoE SFT framework that combines router biases with shared condenser experts.

**Strengths:**

- Novel and Well-Founded Motivation: This paper claims that long-tail of rarely-used experts is "indispensable". This is convincingly supported by the scaling-law analysis in Section 2, which shows that pruning experts (even a small amount) leads to a substantial, non-recoverable performance gap. This provides a solid empirical foundation for the paper's method.

- The Expert Condenser framework is a clever solution to the stated problem. Rather than fighting router collapse with noisy auxiliary losses , the method embraces and induces a "controlled collapse" via bias updates. The idea of designating specific experts to "condense" knowledge from the inactive tail  is an elegant way to preserve information while maintaining sparsity.

**Weaknesses:**

- Unclear Role of "Type-G" Experts: Section 3.2 introduces "Type-G (Ungated) Shared Experts" , which are visualized in Figure 2. These experts are described as standard feed-forward layers applied to every token. Their role is confusing and seems disconnected from the main "condenser" idea, which relates to the "Type-B" experts. It is unclear if these are new parameters added during SFT, and their introduction complicates the paper's claim of not increasing the per-token compute budget.

**Questions:**

Could you clarify the exact selection mechanism for the two Condenser Experts? Based on Appendix M.2 , it appears the correct method is to select the two lowest-bias experts, which aligns with Figure 2 but contradicts the text in Section 3.2.

---

> ### Author Response · Authors · 2025-11-18
>
> Thank you for your valuable review! We would encourage the reviewer to consult our “Response to All Reviewers,” where we list all revisions and additions introduced during rebuttal :)
>
> > Weaknesses 1. Unclear Role of "Type-G" Experts: Section 3.2 introduces "Type-G (Ungated) Shared Experts" , which are visualized in Figure 2. These experts are described as standard feed-forward layers applied to every token. Their role is confusing and seems disconnected from the main "condenser" idea, which relates to the "Type-B" experts. It is unclear if these are new parameters added during SFT, and their introduction complicates the paper's claim of not increasing the per-token compute budget.
>
>
> This is a good point and we’re glad to further clarify this! Yes, the main "condenser" idea is related to the "Type-B" experts, which is also what we proposed in this paper. The "Type-G(ungated) shared experts" are originally from the MoE base model architecture and they are not new parameters added. We have revised section 3.2 to further clarify this and improve the presentation.
>
>
> We also would like to further clarify that our proposed condenser experts are different from traditional Type-G shared experts. Type-G shared experts provide a global residual of shared knowledge as described in DeepSeek-MoE paper. Type-B(Our proposed condenser experts) introduce a gated, constrained backbone that: (1) guarantees a minimal input-adaptive capacity path, improve domain ability, (2) improves router gradient flow, and (3) stabilizes expert utilization—yielding both better specialization and training robustness.  We further **formularize intuitions** of Condenser Experts in Appendix O.
>
> ---
>
> > Questions 1. Could you clarify the exact selection mechanism for the two Condenser Experts? Based on Appendix M.2 , it appears the correct method is to select the two lowest-bias experts, which aligns with Figure 2 but contradicts the text in Section 3.2.
>
> We appreciate the Reviewer for their valuable feedback! The text in section 3.2 is a typo and the condenser experts are selected by identifying the two lowest frequently activated experts. Section 3.2(L222~224) has been revised to correct typo. Thanks!

---

> ### Author Response · Authors · 2025-11-26
>
> Dear Reviewer p9Ex,
>
> Thanks again for the reviews! We deeply appreciate the time and effort you have invested in evaluating our work :)
>
> We've carefully considered your comments and have prepared a detailed rebuttal addressing each point raised. We are hoping reviewers would be able to review our responses and to engage in a discussion before the end of the discussion period. Please let us know if there are any specific areas you'd like us to clarify further or discuss in more detail. We look forward to your thoughts and are eager to engage in a productive dialogue.
>
> Thank you once again for your commitment to the review process.
>
> Best regards,
>
> Authors

---

### Official Review · Reviewer_Fo6M · 2025-11-01

**Soundness:** 2
**Presentation:** 2
**Contribution:** 2
**Rating:** 2
**Confidence:** 3

**Summary:**

This paper presents Expert Condenser, a auxiliary-loss-free framework for post-training Mixture-of-Experts (MoE) large language models. Instead of enforcing balanced expert activation (as in ESFT or DenseMixer), the authors propose bias-driven sparse routing that encourages rarely useful experts to become inactive, while introducing two always-active “condenser experts” that aggregate gradients and knowledge from inactive experts. This aims to preserve long-tail expert knowledge and stabilize routing without adding auxiliary losses. Experiments on DeepSeek, Qwen, OLMoE, and GPT-OSS models show 4–7% accuracy gains over prior methods on math and commonsense reasoning benchmarks.

**Strengths:**

- The results in Table 1/2/3 is clearly outperform other baselines.

- The system efficiency in section 5.3 provide convining support for the proposed methods.

- The experiments covers multiple base models as well as various tasks.

**Weaknesses:**

- The paper lacks ablation studies isolating the individual contributions of different components in the proposed method, such as the bias term and the shared (condenser) experts.

- The idea of shared experts is not new; prior MoE works have already demonstrated their effectiveness. For instance, DeepSeek-MoE explicitly employs shared experts: "Shared Expert Isolation: we isolate certain experts to serve as shared experts that are always activated, aiming at capturing and consolidating common knowledge across varying contexts."

- Section 2 feels unnecessary, as it is natural to me that removing experts would lead to performance degradation. Even dropping less-activated experts can hurt accuracy, as shown in prior works such as https://arxiv.org/pdf/2504.05586.

- Figure 1 contains excessive empty space and would benefit from a more compact and informative redesign.

- The term “base model” in Tables 1–3 is unclear. It appears to refer to results without SFT, but improvements over such a base model are not particularly meaningful. It would be more informative to compare against a vanilla SFT baseline to properly contextualize the gains.

**Questions:**

Please refer to the weakness part.

---

> ### Author Response · Authors · 2025-11-18
>
> Thank you for your valuable review! We would encourage the reviewer to consult our “Response to All Reviewers,” where we list all revisions and additions introduced during rebuttal :)
>
>
> > Weaknesses 1. The paper lacks ablation studies isolating the individual contributions of different components in the proposed method, such as the bias term and the shared (condenser) experts.
>
> We apologize again for not emphazising the appendix content in the main paper (the ablation studies isolating the contributions of adding bias term, removeing auxiliary loss, and extra conder experts are located in Appendix M), which may have caused some confusion. In **Appendix M.1**, we dissect the Condenser Expert algorithm and present empirical results demonstrating that three component
>
> (i) *aux-free only*, which ablated auxiliary losses;
>
> (ii) *aux-free+bias*, which additionally incorporates the bias mechanism
>
> (iii) *aux-free+bias+share*, which further enables expert sharing across tokens
>
> of Condenser Experts contributes to the strong post-training performance. The experiments is conducted on both DeepSeek-V2-Lite (16B) and QWen2-MoE (14B) and tested on 7 general math reasoning downstream datasets.
>
> We also have revised the section 5 of the main paper to emphasize where the ablation studies locate.
>
> ---
> >Weakness 2. The idea of shared experts is not new; prior MoE works have already demonstrated their effectiveness. For instance, DeepSeek-MoE explicitly employs shared experts: "Shared Expert Isolation: we isolate certain experts to serve as shared experts that are always activated, aiming at capturing and consolidating common knowledge across varying contexts."
>
> We appreciate the chance to discuss this further! Our proposed condenser experts are differnt with "shared-experts" in the DeepSeek-MoE. The clarification between previous "shared-experts" in DeepSeek-MoE and our "condenser-experts" can be found in Section-3.2.
>
> Previous shared-experts in DeepSeek-MoE, which is called Type-G (Ungated) Shared Experts in our paper, **always applied and not gated**. They act like an unconditional residual FFN that contributes a fixed function $f_G(x)$ to every token, independent of the router:  $h^{(g)}(x) = \sum_{i=1}^{\hat{n}} \mathrm{FFN}^{(G)}_i(x)$. This provides constant shared capacity but does not interact with the routing dynamics.
>
> Our proposed **condenser-experts**, referred to as **Type-B (Guaranteed–Gated)**, are always selected but still gated on a per-token basis. Two experts  $J_B$ = {$j_1, j_2$}   are guaranteed to appear in the active expert set $S(x)$ for every token and receive router weights $g_{j_b} (x)$. The active set becomes:
>
> $
> S(x) = J_B \cup\ \mathrm{TopK}_{\,i \notin J_B}\big(s_i(x), k-2\big),
> $
>
> and the MoE output is computed as:
> $h^{(b)}(x) = \sum_{i \in S(x)} g_i(x)\,\mathrm{FFN}^{(B)}_i(x)$
>
> **In short,** Type-G (previous shared experts in DeepSeek-MoE), provide a global **residual** of shared knowledge as described in DeepSeek-MoE paper. Type-B(Our proposed condenser experts) introduce a gated, constrained backbone that: (1) guarantees a minimal input-adaptive capacity path, improve domain ability, (2) improves router gradient flow, and (3) stabilizes expert utilization—yielding both better specialization and training robustness. We further **formularize intuitions** of Condenser Experts in Appendix O.
>
> We also provide **empirical evidence** supporting the effectiveness of our proposed condenser experts (Type-B) beyond the functionality of traditional shared experts. In our additional experiments (see response to **Weakness 5**), we compare the SFT baseline — which employs only shared (Type-G) experts — against our model augmented with condenser experts (Type-B). The results consistently show that adding condenser experts leads to substantial improvements across multiple benchmarks.

---

> ### Author Response · Authors · 2025-11-18
>
> >Weakness 3. Section 2 feels unnecessary, as it is natural to me that removing experts would lead to performance degradation. Even dropping less-activated experts can hurt accuracy, as shown in prior works such as https://arxiv.org/pdf/2504.05586.
>
>
>
>
> Thank you for raising this concern. We have cited the [concurrent work](https://arxiv.org/pdf/2504.05586) and we agree that it is intuitively expected that removing experts can lead to performance degradation. However, this intuition alone does not **quantify** *how* performance changes under different compression strategies, nor does it explain *how much* accuracy is lost when removing the least-activated experts. One of the purposes of Section 2 is precisely to provide this quantitative characterization.
>
> The [concurrent work](https://arxiv.org/pdf/2504.05586) cited by the reviewer also observes that pruning less-activated experts can hurt accuracy, but its analysis is limited to *expert pruning within the sparse MoE structure*. In contrast, Section 2 of our paper conducts a more comprehensive empirical study across multiple compression axes: (1) converting MoE models into smaller dense models, (2) pruning experts to create smaller MoE models (the axis explored in the cited work), and (3) reducing the number of activated experts per token. This broader view allows us to analyze not only expert importance, but also how performance scales with the total number of retained expert parameters and the effective activation capacity.
>
> Our extended scaling-law analyses reveal two important findings. First, even though a few “super experts” dominate routing statistics, many rarely activated experts still encode indispensable knowledge: removing the bottom 25% of experts consistently leads to a >10% accuracy degradation. Second, dense conversion and expert pruning exhibit similar sensitivity—that is, both forms of reducing total expert parameters lead to comparable degradation trends. This suggests that the information stored across experts is more distributed than activation frequency alone might imply.
>
>
> These observations naturally motivate our ExpertCondenser design. If rarely activated experts nonetheless contain task-critical information, then simple pruning wastes valuable capacity. Instead, we propose a mechanism that *absorbs* this knowledge into a small set of guaranteed-gated condenser experts, ensuring that information encoded in infrequently used experts is preserved and continuously utilized during post-training. In this sense, Section 2 is not meant to restate an obvious intuition, but to provide the empirical foundation for why consolidating expert knowledge is necessary—and how our method is designed to address this gap.
>
> ---
>
> >Weakness 4. Figure 1 contains excessive empty space and would benefit from a more compact and informative redesign.
>
> We really appreciate the reviewer’s suggestions regarding the presentation in Figure 1. The figure has been adjusted and hope it is more readable now!

---

> ### Author Response · Authors · 2025-11-18
>
> >Weakness 5. The term “base model” in Tables 1–3 is unclear. It appears to refer to results without SFT, but improvements over such a base model are not particularly meaningful. It would be more informative to compare against a vanilla SFT baseline to properly contextualize the gains.
>
> This is a good point! We have added the vanilla SFT baselines in Tab.1~3. We also present the added results below. Our proposed method expertCondenser can consistently outperforms vanilla SFT.
>
> We would also like to mention that we follow the ESFT[2] and DenseMixer[1] to include base model performance in comparison. The purpose of including base model performance is to compare it with post-trained models from different fine-tuning methods, so we can know how much each method can improve the domain performance.
>
> | **Model** | **Model Size** | **Activated #Param** | **Distill Type** | **GPQA Diamond** | **AIME 2024** | **AIME 2025** | **MATH-500** | **AVG** |
> |------------|---------------:|---------------------:|------------------|-----------------:|---------------:|---------------:|---------------:|-----------:|
> | **Qwen3** | **30B** | **3B** | ExpertCondenser (Ours) | 68.8 | 68.3 (82/120) | 51.7 (62/120) | 96.8 | **71.4** |
> |  |  |  | SFT | 58.6 | 63.3 (76/120) | 48.3 (58/120) | 94.8 | **66.3** |
> | **DeepSeek-Coder-V2-Lite** | **16B** | **2.4B** | ExpertCondenser (Ours) | 40.6 | 9.2 (11/120) | 6.7 (8/120) | 68.9 | **31.4** |
> |  |  |  | SFT | 34.2 | 2.5 (3/120) | 2.5 (3/120) | 64.6 | **26.0** |
> | **Qwen2** | **14B** | **2.7B** | ExpertCondenser (Ours) | 34.6 | 6.7 (8/120) | 6.7 (8/120) | 28.6 | **19.5** |
> |  |  |  | SFT | 27.8 | 0.8 (1/120) | 0.0 (0/120) | 20.1 | **12.2** |
>
>
> *Tab.1: Evaluation of vanilla SFT and ExpertCondenser Post-Trained Models Zero-Shot P@ss1:4 Samples Results*
>
>
> | **Model**        | **Model Size** | **Post-train Method** | **BoolQ** | **PIQA** | **SIQA** | **HellaSwag** | **WinoGrande** | **ARC-e** | **ARC-c** | **OBQA** | **AVG** |
> |-------------------|---------------:|-----------------------|-----------|-----------|-----------|---------------|----------------|-----------|-----------|-----------|-----------|
> | **Qwen-2-MoE**    | **14B**        | **SFT**               | 68.8 | 84.7 | 74.5 | 76.8 | 75.6 | 84.6 | 72.8 | 76.4 | **76.8** |
> |                   |                | **ExpertCondenser**   | 72.1 | 84.9 | 75.6 | 81.6 | 79.8 | 88.5 | 78.1 | 84.4 | **80.6** |
> | **OLMoE**         | **7B**         | **SFT**               | 62.5 | 65.8 | 62.8 | 70.7 | 71.4 | 78.4 | 63.7 | 70.6 | **68.2** |
> |                   |                | **ExpertCondenser**   | 66.8 | 79.9 | 72.1 | 80.3 | 78.6 | 86.6 | 70.9 | 75.0 | **76.3** |
>
> *Tab.2: Evaluation of vanilla SFT and ExpertCondenser Post-Trained Models on Commonsense Reasoning Datasets*
>
>
> | **Dataset** | **Model** | **Model Size** | **Activated #Param** | **Post-train Type** | **GSM8K** | **SingleEq** | **SVAMP** | **MultiArith** | **AddSub** | **AQuA** | **mawps** | **AVG** |
> |--------------|------------|---------------:|---------------------:|---------------------|-----------|--------------|-----------|----------------|-------------|-----------|-----------|-----------|
> | **math7k** | **DeepSeek-V2-Lite** | **16B** | **2.4B** | ExpertCondenser (Ours) | 59.4 | 92.5 | 69.1 | 91.5 | 79.5 | 36.1 | 83.6 | **73.1** |
> |  |  |  |  | SFT | 58.4 | 80.6 | 66.2 | 90.2 | 61.0 | 24.8 | 75.8 | **65.3** |
> |  | **QWen2-MoE** | **14B** | **2.7B** | ExpertCondenser (Ours) | 57.2 | 74.6 | 55.7 | 86.0 | 61.8 | 33.1 | 75.6 | **63.4** |
> |  |  |  |  | SFT | 45.8 | 70.2 | 53.6 | 76.2 | 53.3 | 27.6 | 67.8 | **56.4** |
> |  | **OLMoE** | **7B** | **1B** | ExpertCondenser (Ours) | 68.4 | 79.8 | 71.2 | 93.8 | 63.4 | 36.3 | 78.8 | **70.2** |
> |  |  |  |  | SFT | 63.6 | 74.8 | 67.7 | 92.3 | 57.8 | 27.6 | 72.4 | **65.2** |
> | **math14k** | **DeepSeek-V2-Lite** | **16B** | **2.4B** | ExpertCondenser (Ours) | 63.6 | 81.2 | 71.8 | 93.8 | 60.8 | 33.2 | 81.4 | **69.4** |
> |  |  |  |  | SFT | 57.6 | 76.4 | 67.6 | 90.1 | 59.7 | 30.6 | 74.3 | **65.2** |
> |  | **QWen2-MoE** | **14B** | **2.7B** | ExpertCondenser (Ours) | 58.8 | 81.2 | 59.2 | 91.6 | 72.8 | 33.2 | 78.4 | **67.9** |
> |  |  |  |  | SFT | 51.8 | 74.8 | 55.4 | 87.2 | 66.7 | 27.6 | 74.8 | **62.6** |
> |  | **OLMoE** | **7B** | **1B** | ExpertCondenser (Ours) | 67.8 | 81.6 | 72.4 | 86.8 | 68.8 | 32.8 | 79.6 | **70.0** |
> |  |  |  |  | SFT | 65.3 | 76.1 | 67.8 | 81.2 | 64.6 | 31.4 | 72.8 | **65.6** |
>
> *Tab.3: Evaluation of vanilla SFT and ExpertCondenser Post-Trained Models on Math Reasoning Datasets*
>
>
> We sincerely appreciate your insightful feedback.We hope these updates and our responses effectively address your concerns.
>
>
> [1] Improving MoE Post-Training with Precise Router Gradients
>
> [2] Let the Expert Stick to His Last: Expert-Specialized Fine-Tuning for Sparse Architectural Large Language Models

---

> ### Author Response · Authors · 2025-11-26
>
> Dear Reviewer Fo6M,
>
> Thanks again for the reviews! We deeply appreciate the time and effort you have invested in evaluating our work :)
>
> We've carefully considered your comments and have prepared a detailed rebuttal addressing each point raised. We are hoping reviewers would be able to review our responses and to engage in a discussion before the end of the discussion period. Please let us know if there are any specific areas you'd like us to clarify further or discuss in more detail. We look forward to your thoughts and are eager to engage in a productive dialogue.
>
> Thank you once again for your commitment to the review process.
>
> Best regards,
>
> Authors

---

### Official Review · Reviewer_3rWZ · 2025-11-02

**Soundness:** 2
**Presentation:** 2
**Contribution:** 2
**Rating:** 2
**Confidence:** 4

**Summary:**

This paper proposes ExpertCondenser, a training-free auxiliary-loss-free framework for supervised fine-tuning (SFT) of Mixture-of-Experts (MoE) large language models. Its core motivation lies in the overlooked value of rarely activated "long-tail experts" in MoE models—existing methods either ignore these experts or introduce noisy gradients via auxiliary losses. ExpertCondenser reconciles sparse routing and knowledge preservation by leveraging router bias updates to drive inactive experts into dormancy and designating two shared condenser experts to aggregate knowledge from dormant experts. Experiments show that it outperforms state-of-the-art baselines (DenseMixer, ESFT) by over 4% on mathematical reasoning and commonsense QA benchmarks, achieving 2.87× inference speedup and retaining performance under sparse routing.

**Strengths:**

1. Strong compatibility as a post-training module, compatible with various MoE architectures (GPT-OSS, DeepSeek, Qwen, OLMoE) without modifying the core model structure.

2. Balanced performance and efficiency, achieving both performance gains (4%+ on key benchmarks) and computational advantages (2.87× speedup over DenseMixer) via parameter offloading optimization.

**Weaknesses:**

1. I'm not sure how the results reported in the submission are fair. At least, it is fair to re-implement all baselines and fine-tune them with the same dataset. let alone running into tune and finding the best hyperparameters for all methods. The authors motioned in the paper-"We re-implemented ESFT and DenseMixer following the reported setups in (Yao et al., 2025), and reuse their best-reported results when available.", which is contradictory. It is not an apple-to-apple comparison if we reuse results from other papers, as the training datasets are different.

2. The cost of condensing MoE is not new to the community. At least this paper has explored this concept: https://arxiv.org/pdf/2412.00069. Probably, I could find more related works if I searched them entirely.

3. Condensing expert knowledges to certain shared experts may limit the learning capacity of the model, since for different tasks there may be different crucial experts, and the condensing may limit the variability of MoE. Have the authors tested the generalization performance of this method? i.e. training on mathematical reasoning datasets and evaluate on commonsense benchmarks.

4. One of the claims in the paper is that the proposed method achieves better stability in training and expert selection. Could the authors provide supporting results (i.e. training details, loss curves compared with baselines) to further support this claim?


5. Regarding the auxiliary-free training strategy: the selection of experts considers the $\beta$ parameter, but the gating weight does not. Could the authors provide reasons of this design? What would be the results of also using $\beta$ in computing the weights? And how different is the selected experts when using $s_i + \beta_i$ for selection compared to only using $s_i$ (i.e. can be shown with rank metrics with different criteria)?

**Questions:**

If the downstream task domain differs significantly from the pre-training domain (e.g., pre-training on general text but fine-tuning on legal documents), will the knowledge aggregated by condenser experts fail to adapt, leading to performance degradation?

---

> ### Author Response · Authors · 2025-11-18
>
> Thank you for your valuable review! We would encourage the reviewer to consult our “Response to All Reviewers,” where we list all revisions and additions introduced during rebuttal :)
>
> > Weaknesses:
> > 1. I'm not sure how the results reported in the submission are fair. At least, it is fair to re-implement all baselines and fine-tune them with the same dataset. let alone running into tune and finding the best hyperparameters for all methods. The authors motioned in the paper-"We re-implemented ESFT and DenseMixer following the reported setups in (Yao et al., 2025), and reuse their best-reported results when available.", which is contradictory. It is not an apple-to-apple comparison if we reuse results from other papers, as the training datasets are different.
>
> Thank you for pointing out this. To ensure a fully fair comparison, we carefully revisited our experimental setup and have **removed all reused results** in the revised version. The numbers originally reused were the DenseMixer and ESFT–Qwen3 results reported in Table 1 (tune on stanford-S1 dataset) with of our submission. These have now been replaced with our own re-implemented results, trained under the same hyper-parameters as our method. New results are reported below:
>
>
>
> | Model | Model Size | Active #Params | Distill Type | GPQA Diamond | AIME 2024 | AIME 2025 | MATH-500 | AVG |
> |-------|------------|----------------|--------------|--------------|------------|------------|-----------|------|
> | **Qwen3** | **30B** | **3B** | **ExpertCondenser (Ours)** | 68.8 | 68.3 (82/120) | 51.7 (62/120) | 96.8 | **71.4** |
> |        |            |                | DenseMixer | 61.0 | 65.8 (79/120) | 46.7 (56/120) | 95.8 | 67.3 |
> |        |            |                | ESFT | 52.7 | 61.7 (74/120) | 44.2 (53/120) | 92.0 | 62.7 |
> |        |            |                | SFT | 58.6 | 63.3 (76/120) | 48.3 (58/120) | 94.8 | 66.3 |
> |        |            |                | Base Model | 38.9 | 20.8 (25/120) | 7.5 (9/120) | 72.6 | 35.0 |
>
>
> For Tables 1–3, all others baselines results were never reported in previous work, and therefore we re-implemented and fine-tuned all baselines ourselves. This ensures that every method compared in our paper was trained under identical conditions. We verify that models differ only in the fine-tuning method being evaluated, making the comparisons strictly apple-to-apple. For example, for Math7K dataset with DeepSeek-V2-Lite, the hyperparameters used across all methods are:
>
> | Data Type | Optimizer | Learning Rate | Batch Size | Seq. Length | Seed | Warmup Ratio  | LR Scheduler | Attention Impl. | zero_Stage|
> |-----------|-----------|---------------|-------------|--------------|--------------|--------|--------------|------------------| -------|
> | bf16 | AdamW | 1e-5 | 32 | 4096 | 1234 | 0.1 | cosine_with_min_lr |flash_attention_2 | 2 |
>
> All baselines—including ESFT, DenseMixer, SFT, and our ExpertCondenser—were trained with the above settings and identical data splits.

---

> ### Author Response · Authors · 2025-11-18
>
> >Weakness 2. The cost of condensing MoE is not new to the community. At least this paper has explored this concept: https://arxiv.org/pdf/2412.00069. Probably, I could find more related works if I searched them entirely.
>
> Thank you for pointing out the [concurrent paper](https://arxiv.org/pdf/2412.00069) Although the title contains the word “condense,” the motivation of that work is fundamentally different from ours. The cited paper is a pruning-and-compression method: it uses inference on C4 calibration data to estimate expert and layer similarity, and then merges or removes experts to obtain a smaller model with faster inference. Its primary goal is efficiency-driven model compression. In contrast, our work is not a pruning method at all. Our motivation arises from the scaling-law observations in Section 2, which show that rarely activated experts still contain indispensable knowledge. This leads us to ask how MoE models can better preserve and consolidate expert knowledge **during fine-tuning**, rather than discarding it. Our objective is to design a stronger post-training method—not to prune, merge, or compress MoE models.
>
> The architectural mechanisms of the two methods also differ substantially. The cited work eliminates routed experts, averages layer weights based on KL/JS similarity, and replaces selected MoE layers with dense FFNs using fixed gate values computed from calibration data. This removes routing, removes per-token gating, and collapses multiple experts into a single averaged representation. Our method performs none of these operations. ExpertCondenser keeps the full MoE architecture intact and introduces two Type-B Condenser Experts that are always selected yet remain fully gated and fully trainable. Knowledge “condensation” in our framework occurs through gradient-based learning during fine-tuning, not through structural merging or post-hoc weight averaging. Preserving gating is essential: it allows the condenser experts to absorb information dynamically from the entire expert pool, maintaining input adaptivity and avoiding the static behavior introduced by weight-averaged compression methods.
>
> That's say, although both works use the word “condense,” the problems we address, the motivations behind our approaches, and the resulting model architectures are fundamentally different.
>
> ---
> >Weakness 3. Condensing expert knowledges to certain shared experts may limit the learning capacity of the model, since for different tasks there may be different crucial experts, and the condensing may limit the variability of MoE. Have the authors tested the generalization performance of this method? i.e. training on mathematical reasoning datasets and evaluate on commonsense benchmarks.
>
>
> Happy to further discussion on this. Generalization across unrelated tasks is not the primary goal of our work— similar to DenseMixer, our method is specifically designed to improve *post-training performance on the target task*, not to preserve broad multi-domain behavior during SFT. Follow the experimental setting of DenseMixer, we didn't include the experiments to further test the generalization performance.
>
> That said, we conducted an additional experiment to directly evaluate the reviewer’s suggestion. We fine-tuned the OLMoE-7B model on a commonsense dataset and then evaluated the resulting models on mathematical reasoning tasks. Importantly, ExpertCondenser performs comparably to SFT and ESFT in this out-of-distribution setting, and outperforms densemixer. While generalization across unrelated domains is not the focus of our method, the above experiment shows that ExpertCondenser behaves consistently with other fine-tuning methods, without introducing additional degradation.
>
> The results are shown below:
>
> | **FT Dataset** | **Model** | **Model Size** | **Activated #Param** | **Post-train Type** | **GSM8K** | **SingleEq** | **SVAMP** | **MultiArith** | **AddSub** | **AQuA** | **mawps** | **AVG** |
> |---------------|-----------|---------------:|----------------------:|----------------------|-----------|--------------|-----------|----------------|-------------|-----------|-----------|-----------|
> | **commonsense** | **OLMoE** | **7B** | **1B** | Base Model | 16.1 | 23.6 | 17.7 | 9.2 | 21.3 | 22.8 | 13.9 | **17.8** |
> |               |           |                |                      | ExpertCondenser (Ours) | 8.9 | 13.7 | 11.6 | 5.4 | 13.8 | 18.6 | 11.3 | 11.9 |
> |               |           |                |                      | SFT | 6.8 | 14.6 | 12.5 | 5.7 | 13.4 | 17.8 | 8.6 | 11.3 |
> |               |           |                |                      | ESFT       | 11.2 | 15.6 | 13.7 | 4.7 | 12.4 | 16.7 | 7.9 | 11.7 |
> |               |           |                |                      | DenseMixer        | 2.3 | 5.9 | 6.1 | 3.7 | 6.1 | 17.6 | 2.9 | 6.4 |

---

> ### Author Response · Authors · 2025-11-18
>
> >Weakness 4. One of the claims in the paper is that the proposed method achieves better stability in training and expert selection. Could the authors provide supporting results (i.e. training details, loss curves compared with baselines) to further support this claim?
>
>
> Thank you for raising this point. To directly support our claim about improved training and routing stability, we have added a new section in the appendix N with detailed comparisons against ESFT and DenseMixer.
>
> Appendix N reports both the training-loss curves and the gradient-norm trajectories when post-training GPT-OSS on Math7K. For Loss curves, ExpertCondenser converges faster in early training and reaches a lower final loss than both ESFT and DenseMixer, indicating more stable and efficient optimization. For Gradient norms: ExpertCondenser maintains well-bounded, non-spiky gradients. In contrast, DenseMixer exhibits frequent gradient bursts, and ESFT shows early-stage instability.
>
>
> Our theory in Appendix O further shown the stability, to be specific, because condenser experts are always gated, their gate logits $s_j(x)$ participate in the router’s forward and backward passes for every token. Consequently, router parameters $\phi$ receive stable gradients:$\nabla_\phi \mathcal{L}(x) = \sum_{j \in S(x)} \nabla_\phi g_j(x)\,f_j(x),$ where $J_B \subset S(x)$ guarantees non-vanishing gradient flow even for low-probability regions of $s_j(x)$. This acts as a *gradient regularizer*, reducing router collapse and improving routing calibration—analogous to an entropy regularization effect, but achieved structurally through deterministic gating. Besides, every condenser experts expert receives gradient updates for all tokens, preventing long-tail experts from dying and mitigating mode collapse. Ungated shared experts cannot serve this stabilizing function since it bypasses gating.
>
> ---
>
> >Weakness 5. Regarding the auxiliary-free training strategy: the selection of experts considers the $\beta$ parameter, but the gating weight does not. Could the authors provide reasons of this design? What would be the results of also using $\beta$ in computing the weights? And how different is the selected experts when using $\beta+s_i$ for selection compared to only using $s_i$?
>
>
>
> This is a great question! We are more than happy to have further discussion on this:) Our design follows the same principle used in [DeepSeek’s auxiliary-loss-free load-balancing paper](https://arxiv.org/pdf/2408.15664): **the bias term (β or $b_i$) is intentionally applied only to the routing logits before Top-k selection, and not to the gating weights themselves.** This separation is deliberate and has two key reasons.
>
> **(1)** The bias term is meant to influence routing decisions only, not the computation path: In gate's design, the bias-adjusted score is $s_i + b_i$ but the final gating weights $g_{i,t}$ are computed from the unbiased logits $s_i$. As the [Deepseek paper]((https://arxiv.org/pdf/2408.15664)) states in page 4:
> > “the expert bias term $b_i$ is only used to adjust the routing strategy… It is not added to the $g_{i,t}$ that weights the output of the selected experts.”
>
> This separation preserves representation stability. Adding bias to Top-k selection changes which experts are active, but keeps how much each expert contributes (the actual gate weights) consistent with the model’s learned scoring function. If we added $\beta$ into the Softmax/Sigmoid gating weights, the bias would distort the *magnitude* of each expert’s contribution. This would: (1)change the representation distribution layer-by-layer; (2)alter gradients flowing into experts, (3)introduce optimization instability, (4) and disrupt previously learned MoE semantics unnecessarily.
>
>
> **(2)** Adding $\beta$ *after* the matrix multiplication avoids corrupting the learned gating scales:The router logit is: $s_i = G(x^\top e_i)$, Adding $\beta$ before this step (inside the affine transformation) would modify the dot-product geometry, the expert centroids, and the distribution of gating scores. [DeepSeek](https://arxiv.org/pdf/2408.15664) explicitly avoids this because modifying the gating scores themselves introduces “undesired gradients” and destabilizes MoE training. Their paper emphasizes: “Loss-Free Balancing controls load balance without introducing interference gradients.”
>
>
> If $\beta$ also added to the gating weights, it influenced both selection and the weighting, the output would reflect $\beta$-amplified expert contributions, which can saturate softmax/sigmoid and make training unstable. Bias updates would backprop into expert parameters, mixing load-balancing dynamics with LM optimization—exactly the issue DeepSeek warns about.
>
> **Empirically**, DeepSeek also reports that multiplicative or additive bias inside the gating weights results in worse performance ( in [Table 4](https://arxiv.org/pdf/2408.15664)) because the gate distribution becomes unstable.

---

> ### Author Response · Authors · 2025-11-18
>
> >Question 1. If the downstream task domain differs significantly from the pre-training domain (e.g., pre-training on general text but fine-tuning on legal documents), will the knowledge aggregated by condenser experts fail to adapt, leading to performance degradation?
>
> When the downstream task domain differs substantially from the pre-training distribution, any training method may face challenges in out-of-distribution (OOD) generalization. ExpertCondenser is designed specifically to improve **in-domain** post-training performance by preserving and consolidating expert knowledge relevant to the target task. Cross-domain generalization is therefore not the main focus of our method.
>
> Although we do not have the resources to run full MoE pre-training and evaluate OOD behavior in the pre-train → fine-tune pipeline, we conducted an experiment that mirrors the reviewer’s concern within the post-training setting. We fine-tuned OLMoE-7B on a commonsense dataset and then evaluated the resulting models on mathematical reasoning benchmarks—a domain shift that is deliberately large. The results, shown in response to weakness 3 of our rebuttal, indicate substantial degradation for all post-training methods, including SFT, ESFT, DenseMixer, and ExpertCondenser. Importantly, ExpertCondenser does not degrade more than the baselines. Its OOD performance is similar to SFT and ESFT approaches and ourperforms densemixer, suggesting that the condenser mechanism does not cause additional over-specialization or catastrophic forgetting beyond what is typical in domain-specific post-training.

---

> ### Author Response · Authors · 2025-11-26
>
> Dear Reviewer 3rWZ,
>
> Thanks again for the reviews! We deeply appreciate the time and effort you have invested in evaluating our work :)
>
> We've carefully considered your comments and have prepared a detailed rebuttal addressing each point raised. We are hoping reviewers would be able to review our responses and to engage in a discussion before the end of the discussion period. Please let us know if there are any specific areas you'd like us to clarify further or discuss in more detail. We look forward to your thoughts and are eager to engage in a productive dialogue.
>
> Thank you once again for your commitment to the review process.
>
> Best regards,
>
> Authors

---

### Author Response · Authors · 2025-11-18
**To the newly assigned AC and all reviewers**

First, we deeply appreciate the time, expertise, and effort of the AC in evaluating our work. Our paper initially received scores of 6 **[R1 (cRoC)]**, 6 **[R2 (p9Ex)]**, 2 **[R3 (Fo6M)]**, and 2 **[R4 (3rWZ)]**, with corresponding confidences of 4, 2, 3, and 4. We submitted our rebuttal on 18 Nov 2025 at 18:00 EST. As of the review-response lock on 28 Nov, we are still awaiting replies from all four reviewers.

We would also like to thank all reviewers for their thoughtful and valuable feedback. We appreciate that reviewers **[R1, R3, R4]** recognized the strength of our experiments, performance gains, and system efficiency. Reviewers **[R1, R2]** also acknowledged the novelty, well-founded motivation, and the reasonableness of the ExpertCondenser framework as a clever solution for post-training MoE models.

The key contributions of our paper include:

- We conduct a systematic pruning and scaling-law analysis showing that even rarely activated experts contain indispensable knowledge; and removing them significantly degrades performance.

- We extend scaling-law analysis to MoE compression by relating performance to the number of retained expert parameters across dense conversion, expert pruning, and reduced activation budgets.

- We introduce ExpertCondenser, an auxiliary-free post-training framework that enforces sparsity through bias-driven routing while using shared Condenser Experts to preserve knowledge from inactive experts. This approach yields more stable and effective MoE fine-tuning.

We also summarize the key concerns raised and how we have revised the paper to address them:


- **Training stability analysis([R4W4](https://openreview.net/forum?id=DxbLY3Fctc&noteId=On9axbqpSc)):** We added a detailed stability study in ($\color{blue}{\text{Appendix N, Pg. 27}}$), including both loss curves and gradient-norm trajectories.
- **Concerns about reused DenseMixer results ([R4W1](https://openreview.net/forum?id=DxbLY3Fctc&noteId=SyS0oUbwLm))**: We removed reused numbers in the revised version and re-ran results in ($\color{blue}{\text{Section 5.1, Pg.7,  Table 1}}$).
- **Missing vanilla SFT baseline ([R3W5](https://openreview.net/forum?id=DxbLY3Fctc&noteId=Cc5sGQVIqs))**: We added vanilla SFT baselines in ($\color{blue}{\text{Section 5.1-5.2, Pg.7, Tables 1-3}}$) for more complete comparisons.
- **Link between Section 2 and the rest of the paper ([R1W1](https://openreview.net/forum?id=DxbLY3Fctc&noteId=EecZuDE5Rt),[R3W3](https://openreview.net/forum?id=DxbLY3Fctc&noteId=OPBY8hgDan))**: We clarified the motivation and improved the narrative flow in ($\color{blue}{\text{Section 2, Pg.3, L.112-118, L151-155}}$).
- **Generalization and OOD performance([R4W3](https://openreview.net/forum?id=DxbLY3Fctc&noteId=bgusIptdW6), [R4Q1](https://openreview.net/forum?id=DxbLY3Fctc&noteId=MzBoAIJ9Jh))**: Following the reviewer R4’s suggestion, we ran an additional experiment fine-tuning on commonsense and evaluating on math reasoning. The [results](https://openreview.net/forum?id=DxbLY3Fctc&noteId=bgusIptdW6) are available in rebuttal.
- **How condenser experts differ from shared experts([R2W1](https://openreview.net/forum?id=DxbLY3Fctc&noteId=WlrRMFlKlj), [R1Q1](https://openreview.net/forum?id=DxbLY3Fctc&noteId=Dh8Scrg7bi), [R3W2](https://openreview.net/forum?id=DxbLY3Fctc&noteId=AdM0ZXn2yH)) and the theoretical support ([R1W2](https://openreview.net/forum?id=DxbLY3Fctc&noteId=6Zj0NknCvX))**: We revised ($\color{blue}{\text{Section 3.2, Pg.4-5, L.211-228}}$) for clearer explanation, and added a formalized theoretical motivation in ($\color{blue}{\text{Appendix O, Pg.28-29}}$).
- **Difference from pruning-and-compression work.([R4W2](https://openreview.net/forum?id=DxbLY3Fctc&noteId=bgusIptdW6))**: We clarified in the [rebuttal](https://openreview.net/forum?id=DxbLY3Fctc&noteId=bgusIptdW6) that, although both works use the word “condense,” our problem setting, motivation, and architecture are fundamentally different from pruning-based LLM compression.
- **Improved presentation quality ([R3W4](https://openreview.net/forum?id=DxbLY3Fctc&noteId=OPBY8hgDan),[R3W1](https://openreview.net/forum?id=DxbLY3Fctc&noteId=AdM0ZXn2yH))**: We revised ($\color{blue}{\text{Section 2, Pg.3, Fig.1}}$) for clarity and compactness , and improved cross-references to appendices ($\color{blue}{\text{L.286-289, L.214-215}}$).

Additionally, we now support the GPT-OSS-20B model released by OpenAI and fine-tuned it on the Math7K dataset. Results are reported in ($\color{blue}{\text{Section 5.1, Pg.4, Tab.2}}$).

We have carefully addressed **every weakness and question** raised in the reviews, and all discussion points—as well as the new analysis and experiments—have been fully incorporated into the rebuttal and revised version. We sincerely thank the AC and reviewers again for their time and constructive feedback.

---

### Comment · Area_Chair_j9sz · 2025-11-27
**Rebuttal and Discussion Phase**

Dear Reviewers,

Thank you again for your time and effort in reviewing this paper. We are approaching the discussion deadline. I kindly ask you to review the rebuttal and continue the discussion so that we can reach a well-considered decision.

---

### Meta-Review · Area_Chair_KHii · 2026-01-08

**Summary:**

The paper addresses the instability and performance degradation issue during supervised fine-tuning (SFT) of Mixture-of-Experts (MoE) models. The authors first conduct a scaling-law analysis (Section 2) demonstrating that even rarely activated "long-tail" experts contain indispensable knowledge. Based on this, they propose ExpertCondenser, an auxiliary-loss-free framework. It utilizes a bias-driven routing mechanism to enforce sparsity while designating specific "Condenser Experts" (Type-B) that are guaranteed to be selected and gated for every token. These experts aggregate knowledge from the inactive experts without increasing the per-token compute budget. Experiments across several MoE architectures (DeepSeek, Qwen, OLMoE, GPT-OSS) show significant gains in mathematical and commonsense reasoning tasks compared to baselines like DenseMixer and ESFT.

Reviewers initially raised concerns that the original submission reused reported results from other papers for baselines instead of re-implementing them under identical conditions, and the authors re-implemented all baselines from scratch under identical hyperparameters in the rebuttal and demonstrated promising results.

The authors also acknowledged that they exploited the data leak during the ICLR rebuttal stage to identify the reviewers, which was disallowed by ICLR (though this is not a decisive factor in the recommendation).

**Reviewer Concerns:**

The authors addressed most concerns raised by the reviewers in the rebuttal, including the baseline comparisons and the clarifications regarding technical details. The remaining concerns were mainly concentrated on the technical novelty and generalization of the method. Specifically, Reviewers Fo6M and 3rWZ noted that the shared experts idea is not entirely new. In addition, the authors provided new results in their rebuttal that showed that ExpertCondenser performed comparably to SFT and ESFT in out-of-distribution setting, which echoed the concern raised by Reviewer 3rWZ regarding condensing knowledge into certain shared experts limiting the model's total learning capacity. It's also unclear how this SFT-style method will affect the later RL stage in the entire post-training pipeline.

**Reviewer Scores:**

The reviewer scores might slightly increase after the rebuttal, but might still not reach the acceptance threshold.

---

### Decision · Program_Chairs · 2026-01-26

Reject